# PiFlow: Principle-aware Scientific Discovery with Multi-Agent Collaboration

## Abstract

Large Language Model (LLM)-based multi-agent systems (MAS) demonstrate remarkable potential for scientific discovery. Existing approaches, however, often automate scientific discovery using predefined workflows that lack rationality constraints. This often leads to aimless hypothesizing and a failure to consistently link hypotheses with evidence, thereby hindering the systematic reduction of uncertainty. Overcoming these limitations fundamentally requires a principled approach to exploration. We introduce `PiFlow`, an information-theoretical framework, treating automated scientific discovery as a structured uncertainty reduction problem guided by principles (e.g., scientific laws). In evaluations across three distinct scientific domains – discovering nanomaterial structures, bio-molecules, and superconductor candidates with targeted properties – our method significantly improves discovery efficiency, reflected by a 73.55% increase in the Area Under the Curve (AUC) of property values versus exploration steps, and enhances solution quality by 94.06% compared to a vanilla agent system. Overall, `PiFlow` serves as a Plug-and-Play method, establishing a novel paradigm shift in highly efficient automated scientific discovery, paving the way for more robust and accelerated AI-driven research.

## 1 Introduction

Large Language Model (LLM)-based Multi-Agent Systems (MAS) have significantly impacted automated scientific discovery (Minaee et al., 2024; Wang et al., 2023; Zhang et al., 2024c) across a wide range of fundamental fields, including chemistry (Liu et al., 2024; Ghafarollahi & Buehler, 2024a; Yang et al., 2024c; Inoue et al., 2024), biology (Xiao et al., 2024; Nagarajan et al., 2025; Averly et al., 2025; Ghafarollahi & Buehler, 2024b), physics (Jaiswal et al., 2024), and material science (Takahara et al., 2025; Ghafarollahi & Buehler, 2025; Ansari et al., 2024; Wan et al., 2024; Zhang et al., 2024b).

Although proficient in executing experiments within predefined workflows (Lu et al., 2024; Lai & Pu, 2025), these systems often generate hypotheses that lack clear direction, leading to the uncertainty establishing clear links between hypotheses and their supporting or refuting evidence (AI4Science & Quantum, 2023; Zhou et al., 2024; Baek et al., 2024; Schmidgall et al., 2025; Prabhakar et al., 2025; Xie et al., 2023a). Such a disconnect indicates inefficient exploration. Moreover, many of these approaches are tailored for specific tasks, often relying on meticulous prompt engineering that heavily incorporates domain knowledge (as detailed in Appendix D). Consequently, their ability to adapt to new scientific domains is often limited without substantial modifications (Zhang et al., 2025; Kumbhar et al., 2025). These issues culminate in three primary challenges: (a) **aimless hypothesizing**; (b) **unmaintained connections** between hypotheses and evidence during exploration, hindering systematic validation and (c) **limited generalization ability**, where systems effective in one scenario (e.g., material science) often require substantial rework to be applicable in others.

To address these limitations, we introduce `PiFlow`, an information-theoretic framework for structured uncertainty reduction in scientific discovery. Viewing scientific exploration as a game against an unknown and challenging nature, where robust strategies are paramount, `PiFlow` employs Min-Max optimization: **minimizing** cumulative regret for exploitation, while **maximizing** information gain for efficient hypothesis exploration. As a Plug-and-Play module, `PiFlow` integrates with MAS capable of hypothesizing and experimentation. Inspired by the challenge of navigating vast hypothesis spaces,

Figure 1: **Illustration of the potential of a scientific principle in drug discovery.** `PiFlow` directs exploration to prioritize hypotheses aligned with high-potential principles (or their variants), thereby iteratively guiding the discovery towards optimal candidate molecules.

`PiFlow` operates by using fundamental scientific principles, which may be initially proposed by or refined using LLMs.

The iterative selection of principles progressively reduces uncertainty in hypothesizing and the interpretation of evidence, dynamically steering exploration by prioritizing those scientific principles that offer the highest instructive value for continued exploration. Figure 1 illustrates `PiFlow`'s method for assessing and utilizing scientific principles within a drug discovery context.

This principle-aware approach yields systematic information gain: `PiFlow` selects high-potential principles and then guides hypothesizing via three actions, i.e., exploring, validating, or refining their scope and formulation. Thus, `PiFlow` progressively optimizes its guiding scientific principles to effectively steer hypothesizing. Furthermore, leveraging its Min-Max optimization, `PiFlow` theoretically achieves cumulative regret growth of $\mathcal{O}(\sqrt{T})$ over $T$ exploration steps (a detailed proof is provided in Appendix F). This sublinear regret underscores its guaranteed efficiency in navigating complex discovery landscapes. In summary, our contributions are:

(a) We propose a novel paradigm for **principle-aware scientific discovery**, built upon an information-theoretical foundation that offers convergence guarantees.

(b) We develop `PiFlow`, a **Plug-and-Play framework** that seamlessly integrates with existing MAS to enable focused exploration, thereby enhancing discovery efficiency and flexibility.

(c) We conduct **extensive experiments across three distinct scenarios**, demonstrating the broad applicability and significant performance improvements achieved by `PiFlow`.

## 2 RELATED WORK

### 2.1 LANGUAGE MODELS FOR SCIENTIFIC DISCOVERY

Recently, large language models (LLM) have advanced scientific discovery with automation and rational design (Ren et al., 2025; Ma et al., 2024). The internal knowledge of LLMs has demonstrated promising capability in focused chemical and material discovery (Yang et al., 2024c; Zhou et al., 2024; Pu et al., 2024; Ghafarollahi & Buehler, 2024c). While tool-integrated LLMs like SciAgents (Ghafarollahi & Buehler, 2024d), DARWIN (Xie et al., 2023a) and HoneyComb (Zhang et al., 2024a) improve domain-specific reasoning and recall of factual insights, they still struggle to integrate physicochemical laws effectively when refining insights for design, risking biased proposals and inefficient exploration due to inherent hallucinations of LLMs (Zhang et al., 2023b). Human-AI frameworks address this issue by leveraging the knowledge of domain experts (Reddy & Shojaee, 2024; Eythorsson & Clark, 2025), yet remain limited to the scope of hypothesis generation (Alkan et al., 2025), leading to insufficient exploration of complex chemical spaces (Luo et al., 2025). Surveys highlight persistent gaps in efficiency and interpretability (Zhang et al., 2024c; Han et al., 2024), underscoring the need for principled scientific discovery management beyond automated LLM reasoning (Ramos et al., 2024).

### 2.2 APPROACHES OF MULTI-AGENT COLLABORATION

Multi-agent systems (MAS) show promise for complex tasks (Lu et al., 2024; Ghafarollahi & Buehler, 2024d; Ni & Buehler, 2023), yet their application to scientific discovery reveals limitations in current collaboration mechanisms (Tran et al., 2025). Rule-based methods (Zhang et al., 2023a) offer

consistency but their predefined rules lack the flexibility to incorporate nuanced scientific principles (e.g., physicochemical laws) or adapt to unexpected findings, hindering dynamic exploration. The method of AI Researcher (Tang et al., 2025), and role-playing approaches (Tran et al., 2025; He et al., 2024) leverage agent expertise, yet rigid roles can impede adaptation in scientific research (Ramirez-Medina et al., 2025; Lu et al., 2024; Ghafarollahi & Buehler, 2024d), and ensuring collective adherence to scientific principles when interpreting experimental insights is challenging. Model-based methods (Xu et al., 2023; Mu et al., 2023; Li et al., 2023) aim for adaptability by learning from uncertainty, but struggle to build world models that accurately capture complex scientific phenomena and integrate guiding laws (Hao et al., 2023), thereby impairing the balance between information perception and strategic, principle-guided reasoning.

Consequently, existing MAS paradigms often lack a dedicated awareness and systematic application of scientific principles during hypothesis generation and refinement (Gridach et al., 2025; Luo et al., 2025; Reddy & Shojaee, 2024; Su et al., 2024). This highlights a critical need for an explicitly principle-aware multi-agent collaboration framework. Our work addresses this gap, proposing a method where the collaborative discovery process is robustly guided by scientific principles to achieve more efficient and reliable outcomes.

## 3 METHODOLOGY

### 3.1 OVERVIEW

We propose a principle-aware MAS designed to enhance scientific discovery via focused hypothesizing and structured exploration of hypothesis-evidence connections. Figure 2 illustrates the architecture. Its core comprises a Hypothesis-Validation loop that iteratively generates and tests hypotheses. `PiFlow` guides this loop by optimizing accumulated principle-outcome data. Strategic insights dynamically optimized by `PiFlow` are relayed through a Planner agent ($\mathcal{A}_P$) to the Hypothesis Agent within the loop. Subsequent sections will elaborate on `PiFlow`'s specific architecture and its information-theoretical underpinnings.

### 3.2 ARCHITECTURE

Our proposed principle-aware system leverages LLM-based MAS to conduct scientific discovery through strategic hypothesizing. The framework is comprised of two core, interconnected components: (1) an MAS that executes a Hypothesis-Validation loop, and (2) `PiFlow`, which serves as the strategic director for this loop.

**Definition 3.1** (Scientific Principles). The scientific principles are foundational concepts, established laws, or patterns, articulated as natural language statements, that explain phenomena within a specific scientific domain. These principles serve as high-level conceptual building blocks from which specific, testable hypotheses can be derived. This conceptualization aligns with broader discussions on the nature of scientific knowledge (Poincaré, 1906).

**Hypothesis-Validation Loop.** As depicted in the right-hand side of Figure 2, the Hypothesis-Validation loop constitutes the core operational cycle and incorporates two LLM-based agents: Hypothesis Agent ($\mathcal{A}_H$) and Experiment Agent ($\mathcal{A}_E$), detailed below:

(a) **Hypothesis.** Initially, a set of dynamically growing candidate principles $\mathcal{P} = \{p_1, p_2, p_3\}$ (see Definition 3.1) potentially proposed by domain experts (see Appendix P) or extracted by LLMs, is established. In each iteration $t$, $\mathcal{A}_H$ proposes a testable hypothesis, $h_t$, grounded in a selected principle $p_i \in \mathcal{P}$. The proposal is supported by structured rationales (comprising major and minor premises, following Eisape et al. (2023), see Appendix T.2).

(b) **Validation.** Subsequently, $\mathcal{A}_E$ rigorously validates $h_t$ using an experimental tool, denoted $f^*(\cdot)$, which yields a quantitative outcome $y_t = f^*(h_t)$ (e.g., property value of a material). This outcome serves as feedback for subsequent hypothesizing.

This iterative process progressively establishes a record of principle-outcome pairs, $\mathcal{T}_t = \{\langle p_k, y_k \rangle\}_{k=1}^{t}$, linking each hypothesized principle $p_k$ to its observed experimental outcome $y_k$. While this loop systematically generates valuable evidence in trajectory $\mathcal{T}_t$, the selection of potential

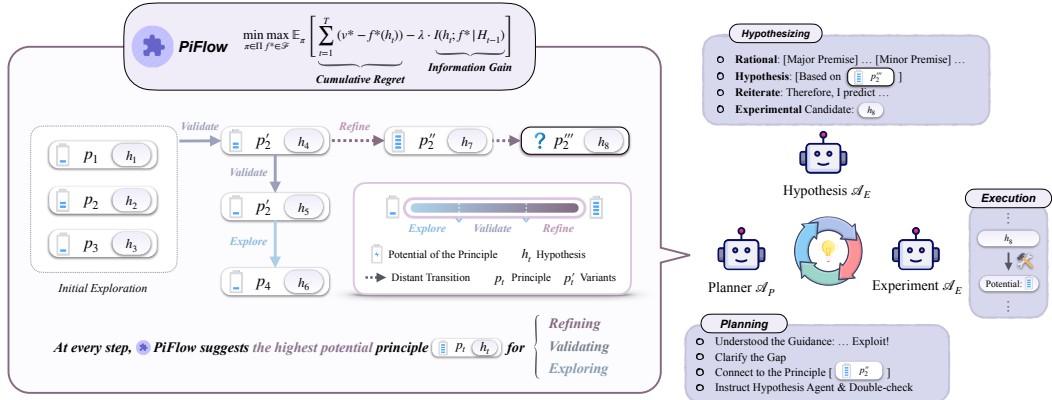

Figure 2: **Overview of the `PiFlow` Architecture for Scientific Discovery.** The `PiFlow` component utilizes Min-Max optimization to strategically select and direct high-potential principles to the Planner agent. The Planner, in turn, guides the Hypothesis-Validation loop, where agents iteratively generate hypotheses $h_t$ from principles $p_t$ at step $t$, execute experiments, and refine understanding. This iterative process is designed to efficiently navigate the discovery landscape.

principles for subsequent hypotheses can lack strategic direction if not externally guided. This may lead to inefficient exploration of the principle space or premature convergence on suboptimal findings.

**Hypothesis steering with `PiFlow`.** To address the potential inefficiencies in principle selection and to instill strategic direction, the `PiFlow` component is introduced. It leverages the dynamically growing set of principle-outcome pairs, $\mathcal{T}_t$ as its primary input. Two steps are conducted to steer the hypothesizing:

(a) **High-potential principle acquisition.** After an initial phase of evidence collection, during which $\mathcal{T}_t$ is populated, `PiFlow` activates its core mechanism: an adversarial Min-Max optimization (detailed in Section 3.3). This optimization process analyzes $\mathcal{T}_t$ to identify a principle, $p^*$ (e.g., $p_2$ in Figure 2), predicted to balance the exploitation and exploration, i.e., have the highest potential, for advancing the scientific inquiry.

(b) **Principle steering for hypothesizing.** Following the identification of the highest-potential principle $p^*$ by the Min-Max optimization, `PiFlow` assigns a potential score to all principles in $\mathcal{P}$. This scoring enables a threshold-based partitioning: principles with high scores (e.g., $p_2'$) drive refinement; those with medium scores trigger further validation ($p_2 \rightarrow p_2'$); and low scores prompt exploration of new conceptual areas.

This closed-loop feedback mechanism ensures that the Hypothesis-Validation cycle is continuously and adaptively steered by strategic insights derived from the system's cumulative experience, as embodied in $\mathcal{T}_t$. We provide a detailed analysis of the distinction from prompt engineering at Appendix E, and an illustrative example at Appendix H.

**Plug-and-Play Modularity of `PiFlow`.** The hypothesis steering mechanism above, which includes `PiFlow` and its interfacing Planner agent $\mathcal{A}_P$, has been intentionally engineered as a modular, Plug-and-Play system. This architectural choice ensures that principle-aware guidance can be readily integrated to enhance MAS engaged in scientific discovery as a part of prompts. As demonstrated in Appendix K, we successfully integrate `PiFlow` with existing ChemToolAgent (Yu et al., 2024) without any architecture modifications. This adaptability positions `PiFlow` as a pivotal enhancement for automated scientific inquiry.

### 3.3 MIN-MAX OPTIMIZATION IN `PiFlow`

In our proposed method, strategic principle selection is achieved through a Min-Max optimization, as presented in Eq. 1. This approach is designed to balance the exploitation of established, high-potential principles (which then guide the formulation of specific hypotheses) with the exploration of novel

ones, while explicitly incorporating information acquisition efficiency:

$$\min_{\pi \in \Pi} \max_{f^* \in \mathcal{F}} \mathbb{E}_\pi \left[ \sum_{t=1}^{T} (v^* - f^*(h_t)) - \lambda \cdot I(h_t; f^* | H_{t-1}) \right] \tag{1}$$

where $\pi \in \Pi$ represents the decision-making policy for selecting principles and $f^* \in \mathcal{F}$ is the evaluation function (e.g., an experimental tool) that characterizes the quantitative outcome yielded from hypothesis $h_t \in \mathcal{H}$, where $\mathcal{H}$ is the hypothesis space. Over $T$ iterations, the policy $\pi$ aims to minimize the objective function in Eq. 1. This objective strategically balances: (1) the summation for cumulative regret, encouraging exploitation of known high-potential principles, and (2) the mutual information, thereby effectively maximizing information gain to foster exploration. This structure allows PiFlow to navigate the complex trade-offs between these two goals.

**Minimizing cumulative fegret (exploitation).** The first term, $\sum_{t=1}^{T}(v^* - f^*(h_t))$, represents the cumulative regret over $T$ iterations. Here, $v^*$ is a theoretical optimal outcome value, and $f^*(h_t)$ is the outcome from hypothesis $h_t$. By minimizing this term, the policy $\pi$ is driven to exploit known high-potential principles and hypotheses to achieve outcomes as close as possible to the optimum $v^*$. This encourages the refinement and validation of promising avenues.

**Maximizing information gain (exploration).** The second term $-\lambda \cdot I(h_t; f^* | H_{t-1})$ promotes exploration. The policy $\pi$ seeks to minimize this term, which is equivalent to maximizing the mutual information $I(h_t; f^* | H_{t-1})$. This mutual information quantifies the expected reduction in uncertainty about the true evaluation function $f^*$ upon observing the outcome of hypothesis $h_t$, given all past observations $H_{t-1} = \{(h_m, y_m)\}_{m=1}^{t-1}$. The trade-off parameter $\lambda > 0$ controls the balance between minimizing regret (exploitation) and maximizing information gain (exploration). A larger $\lambda$ places more emphasis on information acquisition.

**Remark** (The dependencies of $f^*$ in $I(h_t; f^* | H_{t-1})$). The informativeness of a proposed hypothesis $h_t$ is inherently dependent on the nature of the true underlying evaluation function $f^*$. The adversarial nature of the $max_{f^*}$ operator means that the policy $\pi$ must select hypotheses $h_t$ that are expected to be informative even if $f^*$ were to manifest in a way that makes $h_t$ minimally revealing, thus ensuring robustness in information acquisition.

Building upon the theoretical framework above, we derive a computationally tractable algorithm (Algorithm 1) that serves as a principled approximation of the abstract Min-Max objective in Eq. 1. The full rationale and derivation for this approximation are detailed in Appendix G.

In summary, the Min-Max adversarial formulation underpinning PiFlow provides strong theoretical guarantees, notably sublinear regret bounds (formalized in Theorem 1, with proof in Appendix F). Importantly, its operational behavior, which involves practical approximations of this Min-Max solution, demonstrates consistent alignment with these theoretical expectations, as empirically validated in Section 5.4.

**Theorem 1** (Informal). *The Min-Max optimization in PiFlow formulates a trade-off between exploitation (minimizing regret) and exploration (maximizing information gain). Under conditions of finite entropy $H(f^*)$ and bounded evaluation function $f^*$, this optimization provides two key theoretical guarantees: (1) As information gain decreases, the expected regret also decreases; (2) the cumulative regret grows at a sublinear rate of $\mathcal{O}\left(\sqrt{T}\right)$.*

## 4 EXPERIMENT

### 4.1 SETTINGS

To rigorously evaluate the effectiveness and versatility of our PiFlow framework, we conducted experiments across three distinct scientific discovery scenarios. While direct hypothesis validation in real-world labs is prohibitively expensive, we employ high-fidelity surrogate models deployed locally, which serve as the primary evaluation tool. Across all scenarios (Section 4.2), we frame the scientific discovery challenge as a unified task: "*Find a candidate in a complex parameter space that maximizes a target property (e.g., bio-activity).*"

To ensure a fair comparison and focus on core capabilities, all agents utilize QwenMax (Yang et al., 2024a) as the base LLM and are prohibited from accessing external search tools. The complete experimental setup is detailed in Appendix S, and the prompts are provided in Appendix T.

## 4.2 Experimental Scenarios

To comprehensively assess `PiFlow`'s performance, we design three scenarios that represent canonical challenges in scientific exploration: optimization in continuous, discrete, and mixed parameter spaces:

**Nanohelix Optimization (NHO).** We use a surrogate model ($r^2 = 0.98$) trained on DFT-simulated data following Wu et al. (2025) to predict nanohelix chirality from four **continuous** geometric parameters, enabling efficient exploration of the design space (Appendix S.1).

**Molecular Bio-activity Optimization (MBO).** We build a surrogate model ($r^2 = 0.91$) to predict bio-activity from SMILES strings, trained on 50,000 molecules from ChEMBL35 (Zdrazil et al., 2023). This facilitates high-throughput screening in a **discrete** chemical space. (Appendix S.2).

**Superconductor Optimization (SPO).** Following Hamidieh (2018), we train a surrogate model ($r^2 = 0.91$) to map a material's **mixed continuous and discrete** compositional features to its critical temperature ($T_c$), accelerating the discovery of room-temperature superconductors. (Appendix S.3).

## 4.3 Baselines

To evaluate the strategic guidance of `PiFlow` under uncertainty, we therefore benchmark against the following baselines: (1) **Reasoning and Acting (ReAct) (Yao et al., 2022).** ReAct enables agents to iteratively formulate hypotheses, design/execute experiments, and interpret results. It represents a foundational approach to structured reasoning. (2) **Meta Plan Optimization (MPO) (Xiong et al., 2025).** MPO employs a trained LLM planner that provides high-level, general guidance. This serves as a direct counterpoint to `PiFlow`'s explicit, structured mechanism for principled uncertainty reduction. (3) **Vanilla Agent System (Vanilla).** This baseline consists of an MAS operating without any principled strategic oversight. It is intended to establish a performance floor, demonstrating the limitations of unguided exploration reliant solely on agent role-playing. (4) **AI-Researcher** (Tang et al., 2025), a multi-agent research system for autonomous scientific innovation. (5) **The-AI-Scientsit-v2** (Yamada et al., 2025), an end-to-end agentic system that leverages an tree-search methodology for prompting research progress. We adapt these two AI-Scientist methods into our tasks by excluding literature review and manuscript drafting for a fair comparison.

While other advanced frameworks exist, such as Agent-Oriented Planning (AOP) (Li et al., 2024) and Reason for Future, Act for Now (RAFA) (Liu et al., 2023), their objectives diverge from the scope of de novo evidence-gathering process central to `PiFlow`. Therefore, our selection of baselines is specifically designed to isolate and evaluate our contribution to discovery efficiency in uncertain environments.

## 4.4 Evaluation Metrics

Two metrics are employed to evaluate the performance of `PiFlow`, as detailed below:

**Solution Quality (SQ).** We measure the optimal objective value with a percentage relative to the theoretical maximum value $\mu_{\text{absolute}}$, denoted as:

$$\text{SQ} = \frac{\max\{y_k \mid \langle p_k, y_k \rangle \in \mathcal{T}_t\}}{\mu_{\text{absolute}}} \times 100\%. \qquad (2)$$

Specifically, for NHO, the theoretical maximum is reported as $\mu_{\text{absolute}}^{\text{g-factor}} = 2.0$ following Greenfield et al. (2021). The larger the g-factor, the stronger the chirality. For MBO, a strict threshold of $\mu_{\text{absolute}}^{\text{pchembl}} = 6.5$ has been reported (Lenselink et al., 2017) for lead compound optimization stage, larger value indicates a higher bio-activity and vice versa. For SPO, the reference value is set to be $25°C$, which is equal to room temperature 298.15K, denoted as $\mu_{\text{absolute}}^{\text{Tc}} = 298.15K$.

**Area Under the Curve (AUC).** Exploration efficiency requires both (1) **rapid convergence** and (2) **high objective values**. We quantify these two factors by defining the AUC metric. Given the

Table 1: Comparisons between `PiFlow` and baselines.

| Method | NHO (g-factor) | | MBO (pChEMBL) | | SPO ($T_c$) | |
|---|---|---|---|---|---|---|
| | **AUC (%)** ↑ | **SQ** (%) ↑ | **AUC** (%) ↑ | **SQ** (%) ↑ | **AUC**(%) ↑ | **SQ** (%) ↑ |
| ReAct | 35.85 ±7.12 | 41.96 ±0.82 | 29.61 ±9.74 | 43.11 ±9.98 | 5.29 ±0.69 | 6.41 ±0.96 |
| Vanilla | 35.96 ±22.38 | 46.76 ±7.29 | 29.71 ±12.66 | 49.22 ±8.30 | 11.39 ±11.33 | 14.16 ±13.37 |
| MPO | 43.99 ±2.79 | 51.29 ±7.77 | 31.18 ±9.12 | 57.28 ±5.85 | 12.68 ±7.75 | 33.20 ±23.75 |
| AI-Researcher | 46.45 ±1.98 | 53.12 ±0.87 | 42.72 ±12.71 | **95.66** ±6.08 | 16.36 ±2.22 | 25.69 ±2.64 |
| The-AI-Scientist-v2 | 49.27 ±1.51 | 56.67 ±4.73 | 36.32 ±8.67 | 62.47 ±6.55 | **28.43** ±2.55 | 29.85 ±2.18 |
| **PiFlow** (ours) | **63.51** ±11.18 | **76.82** ±4.54 | **64.57** ±23.65 | 84.55 ±29.63 | 21.51 ±2.80 | **34.85** ±1.19 |

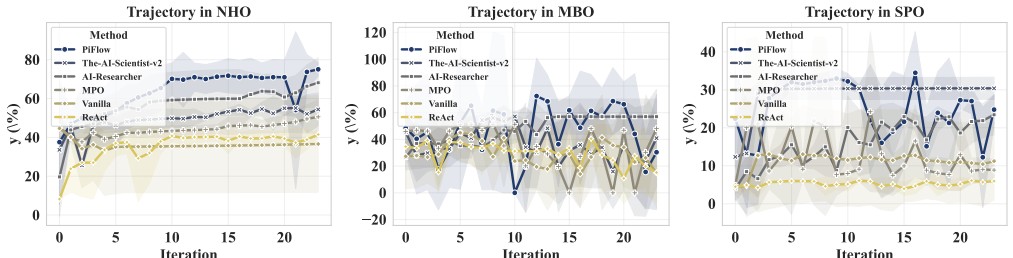

Figure 3: Trajectory comparisons for different optimization methods.

trajectory $\langle p_k, y_k \rangle \in \mathcal{T}_t$ across $t$ steps, AUC is computed via the trapezoidal rule. For meaningful comparisons, we normalize by the maximum possible area:

$$\text{AUC} = \frac{\sum_{i=1}^{t-1} \frac{y_i + y_{i+1}}{2}}{\mu_{\text{absolute}} \cdot (t-1)} \times 100\% \tag{3}$$

In summary, SQ measures the quality of the final outcome, while AUC evaluates the entire discovery process by rewarding both speed and consistency. The detailed rationale for these metric designs is provided in Appendix R.2.

## 5 RESULTS

### 5.1 PERFORMANCE COMPARISON

We use SQ to compare the overall capability of reaching the objective solution, and AUC to assess the efficiency, reflecting progress towards better outcomes over the exploration process.

Table 1 shows that `PiFlow` consistently outperforms traditional baselines (ReAct, Vanilla, MPO) by wide margins across all tasks. Compared to SOTA agents, `PiFlow` demonstrates superior global search capabilities.

As visualized in Figure 3, while The-AI-Scientist-v2 shows high AUC in SPO, its trajectory quickly stagnates, indicating entrapment in local optima. In contrast, **`PiFlow` exhibits oscillatory trajectories, reflecting an active exploration strategy** that effectively escapes local traps. This dynamic behavior allows `PiFlow` to discover superior final solutions (highest SQ in NHO and SPO) and maintain high efficiency (competitive AUC in MBO), verifying that the observed variance stems from productive exploration rather than instability. Further details regarding baseline analysis and **statistical significance tests** for the AUC results are provided in Appendix I.

> **Takeaway:** `PiFlow` achieves state-of-the-art results by prioritizing active exploration over passive exploitation. Trajectory analysis reveals that unlike baselines which suffer from premature stagnation, `PiFlow` dynamically navigates the solution space to locate better global optima.

### 5.2 ABLATION STUDY

We conduct several ablations to evaluate `PiFlow`. For these studies, conducted on the NHO task, performance is evaluated based on the same metrics, AUC (%) and SQ (%).

**Plug-and-Play.** To isolate the direct benefit of `PiFlow`, we compared the performance with two different LLMs, GPT4.1-mini and Qwen3-32B under the setting of `w/` and `w/o` `PiFlow`. This is

achieved by **only including or excluding** the steered principle to the Planner Agent via prompt, which then directs subsequent hypothesizing and validation.

As shown in Table 2, for GPT4.1-mini, integrating `PiFlow` increases the AUC from 37.12% to 41.68% and substantially boosted the SQ from 40.14% to 66.38%. This represents an approximate 12.3% improvement in AUC and a significant 65.4% improvement in SQ. Similarly, for the Qwen3-32B model, the inclusion of `PiFlow` improves AUC from 27.04% to 37.51% (a 38.7% increase) and SQ from 54.84% to 58.76% (a 7.1% increase).

**Thought Mode Effect.** We also conduct ablations on the internal thought mode of the LLM (referred to as Think in Table 3). This is for agents based on Qwen3-32B and Qwen3-8B models, which support ON/OFF `<think>...</think>` generation with system prompt including or excluding `/no_think`. This mode is intended to enable more explicit reasoning steps.

Table 2: Ablation Study with/without `PiFlow`

| Method/Setting | | AUC (%) | SQ (%) |
|---|---|---|---|
| GPT4.1-mini | w/ `PiFlow` | **41.68** $\pm 17.91$ | **66.38** $\pm 14.90$ |
| | w/o `PiFlow` | 37.12 $\pm 16.04$ | 40.14 $\pm 13.97$ |
| Qwen3-32B | w/ `PiFlow` | **37.51** $\pm 7.70$ | **58.76** $\pm 6.18$ |
| | w/o `PiFlow` | 27.04 $\pm 21.50$ | 54.84 $\pm 24.41$ |

Interestingly, for both Qwen3-32B and Qwen3-8B, disabling the Thought Mode leads to improved performance, as shown at Table 3. We hypothesize that this phenomenon is due to *cognitive fixation*. As the key issue is, how do LLM propose scientific hypothesis, forcing the LLM to generate an explicit Chain-of-Thought may cause it to commit prematurely to its own initial line of

Table 3: Effect of Thinking.

| Method/Setting | | AUC (%) | SQ (%) |
|---|---|---|---|
| Qwen3-32B | w/ Think | 37.51 $\pm 7.70$ | 58.76 $\pm 6.18$ |
| | w/o Think | **45.51** $\pm 11.19$ | **68.86** $\pm 17.67$ |
| Qwen3-8B | w/ Think | 30.55 $\pm 27.45$ | 54.49 $\pm 22.25$ |
| | w/o Think | **42.09** $\pm 16.55$ | **61.59** $\pm 16.43$ |

reasoning. If the first step in its logic is flawed, the entire chain can be led astray, creating a cognitive fixation that is hard to escape. In contrast, disabling the think mode may force the model to rely more on its powerful, holistic pattern-matching capabilities, allowing it to make more intuitive leaps directly from the data (`PiFlow`'s guidance and the experimental history) to a hypothesis, bypassing potentially flawed intermediate reasoning steps.

> **Takeaway:** Our ablations confirm `PiFlow` is a robust, Plug-and-Play enhancement that consistently boosts performance across models. Concurrently, we find that an agent's internal reasoning is critical. This highlights the synergy between high-level strategic guidance from `PiFlow` and internal reasoning.

### 5.3 FURTHER ANALYSES OF ROBUSTNESS

We conducted an extensive suite of **eight experiments** to rigorously evaluate the robustness and multifaceted utility of `PiFlow`. While full experimental details are provided in the Appendix, we highlight the following critical findings:

**PiFlow can recover from poor initialization.** The system demonstrates resilience by successfully recovering from deliberately incorrect initial principles, demonstrating that `PiFlow` does not rely on the initial principles. This underscores the robustness of the Min-Max (Appendix P).

**Temporal dynamics of principle evaluation.** `PiFlow` continuously re-evaluates the utility of scientific principles as new evidence is gathered. This allows it to dynamically discard initially promising but ultimately flawed avenues while elevating principles that prove more fruitful later, showcasing an adaptive balance between exploration and exploitation (Appendix N).

**Superior performance against numerical search.** On complex tasks with ill-defined search spaces, `PiFlow` substantially outperforms Bayesian Optimization, achieving an SQ of 84.55% versus BO's 38.15% in the MBO task, without requiring manual parameter space engineering (Appendix J).

**Seamless Plug-and-Play integration.** We validate the practicality of `PiFlow` through a successful integration with ChemToolAgent (Yu et al., 2024), guiding it to a high-value molecule (pChEMBL of 5.90) without requiring any architectural modifications (Appendix K).

**Manageable computational complexity.** The core decision mechanism of `PiFlow` has a computational complexity of $\mathcal{O}(t^2 \cdot d)$, where $t = |\mathcal{T}_t|$ and $d$ is the embedding dimension. We also evaluate the actual running time of `PiFlow` and confirm its high efficiency (Appendix L).

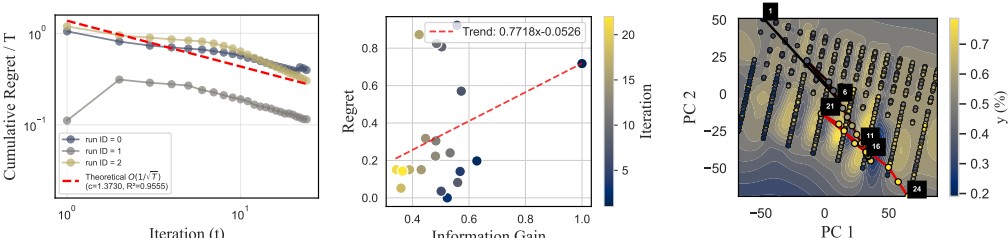

(a) Average Regret Dynamics  (b) Regret vs. Information Gain  (c) Exploration Trajectory

Figure 4: Empirical Validation of `PiFlow`'s Theoretical Alignment.

**Cost-effectiveness.** `PiFlow`-MAS achieves a 27% reduction in token consumption and a 5.6x speedup compared to the Vanilla Agent. Crucially, `PiFlow` module only accounts for 1.5% of the total token cost (Appendix M).

**Generalizability and controllability.** `PiFlow` is compatible with various LLM backbones (Appendices O), and its behavior can be tuned via $\lambda$, for which we provide a clear heuristic (Appendix Q).

## 5.4 THEORETICAL ALIGNMENT

To empirically validate the theoretical guarantees (Theorem 1) of `PiFlow`, we analyze key aspects of its exploration dynamics, including (1) the bound of average regret and (2) the relationship between regret and information gain, as shown in Figure 4.

**(Theoretical Prediction 1) Average regret decay with $\mathcal{O}\left(\frac{1}{\sqrt{T}}\right)$.** Figure 4a presents the average regret as a function of iterations on a log-log scale. The alignment evidenced by average regret trajectories adhering to the $T^{-0.5}$ decay (fitted $c \cdot T^{-0.5}$ with $c = 1.37, r^2 = 0.96$), a pattern most runs consistently follow. Notably, one run (ID=1), after an initial sharp regret decrease, shows a transient increase before resuming decay. This illustrates `PiFlow`'s robust exploration avoiding potential local optima with a characteristic of its Min-Max strategy.

**(Theoretical Prediction 2) As information gain decreases, the expected regret also decreases.** As shown in Figure 4b, the scatter plot of regret versus information gain (points colored by iteration) reveals a clear positive association, confirmed by a fitted trend line: lower information gain generally corresponds to lower regret. Early iterations (higher information gain and regret) transition to later iterations (lower information gain and regret).

**Theoretical alignments in agent trajectory dynamics.** Figure 4c displays the trajectory of `PiFlow` on the NHO objective landscape, which is visualized via Principal Component Analysis (PCA) with contours indicating g-factor values. The path demonstrates a principled strategy: initial broad exploration (iter 1-16), followed by navigating a low-quality "valley" to escape a local optimum (iter 16-21), and culminating in efficient convergence to a high-g-factor region (iter 21-24).

> **Takeaway:** Our empirical analysis corroborates the theoretical guarantees of `PiFlow`, demonstrating both theoretically alignment and effectiveness.

## 6 CONCLUSION

In conclusion, we propose `PiFlow`, a Plug-and-Play module to strategically guide the Hypothesis-Validation loop through a steering mechanism to address challenges of aimless hypothesizing and unclear connections between hypotheses and evidence. Our approach utilizes a Min-Max optimization that explicitly balances exploitation of high-potential principles with exploration of novel hypotheses, guaranteed by a sublinear average regret bound. Extensive experiments demonstrate that `PiFlow` can adaptively navigate complex hypothesis spaces without premature convergence on suboptimal solutions, yielding significant improvements over baselines. A detailed discussion of its current limitations and future potential is provided in Appendix C. We hope `PiFlow` can contribute to advancing automated scientific discovery, inspiring further exploration and innovation.

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

CONTENTS

## A    USE OF LARGE LANGUAGE MODELS

We utilized a large language model to assist with proofreading and polishing the language in this manuscript.

## B    BROADER IMPACT

`PiFlow` addresses critical bottlenecks in AI-driven scientific discovery where uncertainty leads to aimless exploration. Our method innovatively frames discovery as a structured uncertainty reduction problem. By using an information-theoretic approach within a hypothesis-validation loop, `PiFlow` systematically filters for instructive scientific principles. Ultimately, `PiFlow` establishes a new paradigm for automated research, enabling more targeted exploration and accelerating the generation of impactful scientific insights. In Materials Discovery, it speeds the identification of novel compounds like advanced nanomaterials or superconductors, as shown in our tasks. For Biological Discovery, it enhances the search for effective molecules and the understanding of complex systems. Its principles promise similar advancements in other data-intensive fields, from chemistry to medical sciences, facing vast and uncertain hypothesis spaces.

## C    LIMITATION AND FUTURE WORK

While `PiFlow` shows notable improvements through its principled Min-Max optimization, its practical implementation approximates a key theoretical component. This means the current system may not fully capture all nuances of true, model-based information gain, especially the direct adversarial interplay with all possible manifestations of the unknown evaluation function $f^*$ from the theoretical objective.

Future research could explore more direct estimations of mutual information for this heuristic within the `PiFlow` framework to potentially further enhance its strategic guidance. Furthermore, we observed that disabling the LLM's "Thought Mode" surprisingly improves performance, suggesting that forced Chain-of-Thought can induce *cognitive fixation*. This finding motivates the development of more flexible reasoning frameworks for agents, aiming to better balance deliberate logic with intuitive generation.

## D    DETAILED METHODOLOGICAL REVIEW

A hallmark and fundamental limitation of many contemporary LLM-based agent systems in scientific discovery is their limited generalizability. As detailed in Table 4, these frameworks are often characterized by a **tight coupling between their core logic and a specific scientific domain**. This means their implementations, tool integrations, and most critically, their prompt engineering strategies are meticulously tailored for a single scenario, such as organic chemistry or materials science. Although systems like The AI Scientist (Lu et al., 2024) and Agent Laboratory (Schmidgall et al., 2025) demonstrate strong capabilities in scientific research, they are specifically designed for AI domain, along with a focus of the whole workflow rather than strategic decision-making problem. Consequently, transferring these systems to a new domain necessitates significant re-engineering, restricting their out-of-the-box applicability and hindering the development of truly universal systems.

In contrast, our `PiFlow` is designed to overcome this challenge by **decoupling the strategic decision-making layer from the task execution layer**. By architecting `PiFlow` as a domain-agnostic, Plug-and-Play module, it provides strategic guidance to a minimal hypothesis-testing MAS without embedding domain-specific knowledge within its own logic. This architectural choice yields superior flexibility and obviates the need for extensive, domain-specific prompt engineering for the strategic component, enabling seamless adaptation across diverse scientific fields.

> **Take-away**: `PiFlow` introduces a domain-agnostic, plug-and-play architecture by **decoupling** strategic decision-making from domain-specific execution. This design overcomes the critical generalizability limitations inherent in tightly-coupled scientific agent systems.

Table 4: Comparison of Generalization Limitations in LLM-Agent Systems

| Method Name | Core Implementation | Adaptability |
|---|---|---|
| The AI Scientist (Lu et al., 2024) | Generating novel research ideas, writes code, executes experiments, visualizes results, describes its findings by writing a full scientific paper | AI research |
| Agent Laboratory (Schmidgall et al., 2025) | Accepting a human-provided research idea and progressing through three stages – literature review, experimentation, and report writing to produce comprehensive research outputs, including a code repository and a research report | AI research |
| CellAgent (Xiao et al., 2024) | Constructing LLM-driven biological expert roles - planner, executor, and evaluator - each with specific responsibilities | scRNA-seq Analysis |
| DrugAgent (Liu et al., 2024) | Employing an LLM Planner that formulates high-level ideas and an LLM Instructor that identifies and integrates domain knowledge when implementing those ideas | Drug discovery |
| IAN (Nagarajan et al., 2025) | Leveraging popular pathway and regulatory datasets for protein-protein interactions to perform analysis through a LLM-based multi-agent architecture | Protein-protein interactions analysis |
| (Ghafarollahi & Buehler, 2025) | Automating alloy design and discovery using physics-aware multimodal multi-agent AI | Alloy design and discovery |
| (Takahara et al., 2025) | Accelerating inorganic materials design using generative AI agents. | Inorganic materials design |
| LIDDIA (Averly et al., 2025) | Language-based intelligent agent for drug discovery tasks. | Drug discovery |
| dZiner (Ansari et al., 2024) | Rational inverse design of materials facilitated by AI agents. | Inverse design of materials |
| MOOSE-Chem (Yang et al., 2024c) | Utilizing Large Language Models for rediscovering unseen chemistry scientific hypotheses. | Chemistry hypothesis discovery |
| OmniScience (Prabhakar et al., 2025) | Domain adaptive pretraining, instruction tuning and reasoning-based knowledge distillation | Scientific reasoning |
| (Zhang et al., 2024b) | Large Language Model-Based AI Agent for research in organic semiconductor devices | Organic semiconductor device research |
| DrugAgent (Inoue et al., 2024) | Explainable drug repurposing agent with Large Language Model-based reasoning | Drug repurposing |
| ProtAgents (Ghafarollahi & Buehler, 2024b) | Protein discovery via large language model multi-agent collaborations combining physics and machine learning | Protein discovery |
| **PiFlow (ours)** | A unified framework for optimizing and searching-oriented scientific discovery problems | Board area in science |

## E  Distinction from Advanced Prompt Engineering

We define a scientific principle following Definition 3.1. Formally, each principle is represented as a structured text proposition that can be algorithmically scored by our MinMax optimization (as exampled in Figure 1). Unlike arbitrary text strings, these principles is supposed to be with logical consistency, and this structural requirement is what distinguishes them from general prompts. In short, principles represent foundational concepts or established patterns within a domain (e.g., "higher hydrophobicity often correlates with better cell membrane penetration"). These "principles", whether has been validated or not, can be proposed by experts or, crucially, extracted from the LLM's own vast pre-trained knowledge. They serve as high-level starting points to generate specific, testable hypotheses.

The core distinction of our `PiFlow` from prompt engineering lies not in the format of the guidance (which is text), but in the algorithmic generation of that guidance:

- **Prompt engineering baseline.** The agent quickly became trapped in local optima. Its process was highly repetitive (e.g., repeatedly stating "The system's behavior is governed by the interplay..."), and it eventually stagnated, making only meaningless tweaks to the g-factor (e.g., $1.1012 \rightarrow \cdots \rightarrow 1.1030$). This demonstrates the limitation of a static, unguided hypothesis-testing loop.

- **Systematic breakthroughs with `PiFlow`.** In contrast, the PiFlow-guided agent demonstrated structured, cumulative learning:

  1. **(Early 1 3 iterations) Principled exploration**: It begins with diverse hypotheses to maximize information gain.
  2. **(Medium stage) Discovery**: It identified non-monotonic relationships (e.g., "deviations beyond an optimal configuration reduce chirality"), forming a natural language-based identification.
  3. **(Late stage) Paradigm Shift**: Ultimately, it synthesized the principle of "minimal radius + maximal turns + optimized pitch", causing a decisive shift in the search space from (fiber_radius=40, helix_radius=70) to (fiber_radius=20, helix_radius=20) and identifying the non-linear sensitivity of the pitch parameter. This unlocked the significant g-factor improvement ($0.84 \rightarrow 1.28 \rightarrow 1.41 \rightarrow 1.51$).

The core of `PiFlow` is to force the LLM to structure its disorganized internal knowledge into explicit, falsifiable hypotheses. The `PiFlow` then uses quantitative feedback from experiments to iteratively refine its understanding regarding the task. The value is not in feeding the LLM new knowledge, but in providing a strategic framework to systematically test and organize the knowledge it already possesses, yielding a loop of *LLM Knowledge → External Evidence → Guided LLM Action*, rather than *LLM → LLM*.

In summary, the algorithmic core lies in Equation 1 and Algorithm 1: the Min-Max optimization systematically balances regret minimization with information gain maximization. This is operationalized through **dynamic principle scoring that updates based on accumulated evidence** $\mathcal{T}t = \langle p_k, y_k \rangle k = 1^t$. Advanced prompt engineering lacks this principled mathematical framework for evidence integration and strategic trade-offs.

> **Take-away**: Instead of relying on static prompts that lead to local optima, `PiFlow` employs a Min-Max optimization to algorithmically generate and refine structured, falsifiable principles. By systematically integrating experimental feedback, it transforms the LLM's latent knowledge into a dynamic, self-correcting engine for scientific discovery, enabling cumulative learning and strategic breakthroughs.

## F  Proof of the Theorem

We recall the elements here and formally proof the convergence along with boundary of the system. Here we denote $\pi$ is the language model policy from policy space $\Pi$, $f^*$ is the acquisition function from function space $\mathcal{F}$, $h_t$ is the hypothesis at time step $t$, $H_{t-1} = \{h_1, h_2, \ldots, h_{t-1}\}$ is the history

of hypotheses, $v^*$ is the SQ achievable by any hypothesis, and $I(h_t; f^* \mid H_{t-1})$ is the conditional mutual information.

According to the original formulation (Eq. 1), the cumulative regret can be expressed as

$$R_T(\pi, f^*) = \mathbb{E}_\pi \left[ \sum_{t=1}^{T} (v^* - f^*(h_t)) \right]$$

and the cumulative information gain is

$$IG_T(\pi, f^*) = \mathbb{E}_\pi \left[ \sum_{t=1}^{T} I(h_t; f^* | H_{t-1}) \right]$$

**Information gain approaches zero.** With information theory, the mutual information can be written as:

$$I(h_t; f^* | H_{t-1}) = H(f^* | H_{t-1}) - H(f^* | H_{t-1}, h_t)$$

A critical property for convergence is that the total information gain is bounded,

$$\sum_{t=1}^{\infty} I(h_t; f^* \mid H_{t-1}) \leq H(f^*) < \infty$$

This follows from the chain rule of mutual information,

$$\sum_{t=1}^{T} I(h_t; f^* \mid H_{t-1}) = I(H_T; f^*) \leq H(f^*)$$

Since the entropy $H(f^*)$ is finite, the cumulative information gain is bounded regardless of how many steps $T$ we take. This implies that:

$$\lim_{t \to \infty} I(h_t; f^* \mid H_{t-1}) = 0.$$

In other words, the marginal information gained from each new hypothesis must eventually approach zero.

**As information gain decreases, the expected regret also decreases.** As we mentioned before, the regret $R_T(\pi, f^*)$ is defined as:

$$R_T(\pi, f^*) = \mathbb{E}_T[v^* - f^*(h_t)].$$

Now apply Jensen's Inequality, let $X = v^* - f^*(h_t)$, we have

$$\mathbb{E}[X^2 \mid H_{t-1}] = \mathbb{E}[(v^* - f^*(h_t))^2 \mid H_{t-1}].$$

Use $\phi(x) = x^2$, as it is convex, giving

$$(\mathbb{E}[X \mid H_{t-1}])^2 \leq \mathbb{E}[X^2 \mid H_{t-1}],$$

take square roots:

$$\mathbb{E}[v^* - f^*(h_t) \mid H_{t-1}] \leq \sqrt{\mathbb{E}[(v^* - f^*(h_t))^2 \mid H_{t-1}]}.$$

Now we deal with the second moment of the regret. As the $v^*$ is constant, therefore, with the variance formula, we have

$$\mathbb{E}\left[(v^* - f^*(h_t))^2 \mid H_{t-1}\right] = \mathrm{Var}(f^*(h_t) \mid H_{t-1}) + (v^* - \mathbb{E}[f^*(h_t) \mid H_{t-1}])^2$$

Both terms (variance and bias) are non-negative, and our goal is to bound this expression. The variance term $\text{Var}(f^*(h_t) \mid H_{t-1})$ captures the uncertainty in $f^*(h_t)$ given the history.

Since the information theory proofed that, the variance of a function can be bounded by mutual information, akin to entropy bounds $H(f^*) \leq \log(|\mathcal{F}|)$, for the first term, we have

$$\text{Var}(f^*(h_t) \mid H_{t-1}) \leq c \cdot I(h_t; f^* \mid H_{t-1}),$$

where $c$ is constant that depends on the range of $f^*$, this follows because the mutual information bounds the expected variance of conditional expectations.

Since we can direct view the $f^*(h_t)$ as a random variable over the joint distribution of $f^*$ and $h_t$, we can apply a concentration inequality. A standard result in information-directed sampling states that for a bounded random variable like $f^*(h_t)$ here, the second moment of the regret $(v^* - f^*(h_t))$ can be bounded as:

$$\mathbb{E}\left[(v^* - f^*(h_t))^2 \mid H_{t-1}\right] \leq c \cdot I(h_t; f^* \mid H_{t-1}),$$

where the constant $c$ depends on $|\mathcal{F}|$.

Finally, we get the inequality of $R_T(\pi, f^*)$:

$$R_T(\pi, f^*) \leq \sqrt{\mathbb{E}_T\left[(v^* - f^*(h_t))^2 \mid H_{t-1}\right]} \leq c \cdot \sqrt{I(h_t; f^* \mid H_{t-1})}$$

This inequality demonstrates that as the information gain $I(h_t; f^* \mid H_{t-1})$ decreases, the expected regret also diminishes.

**The cumulative regret grows at a rate $O(\sqrt{T})$.** Based on the inequality in the last step and to find the cumulative regret bound, we need to sum this inequality over all time steps:

$$\sum_{t=1}^{T} R_t(\pi, f^*) \leq c \cdot \sum_{t=1}^{T} \sqrt{I(h_t; f^*|H_{t-1})}$$

Applying the Cauchy-Schwartz inequality:

$$\sum_{t=1}^{T} \sqrt{I(h_t; f^*|H_{t-1})} \leq \sqrt{T \cdot \sum_{t=1}^{T} I(h_t; f^*|H_{t-1})}$$

Since we've already established that the total information gain is bounded:

$$\sum_{t=1}^{T} I(h_t; f^*|H_{t-1}) \leq H(f^*)$$

We can substitute this bound:

$$\sum_{t=1}^{T} \sqrt{I(h_t; f^*|H_{t-1})} \leq \sqrt{T \cdot H(f^*)}$$

Therefore, the cumulative regret is bounded by:

$$\sum_{t=1}^{T} R_t(\pi, f^*) \leq c \cdot \sqrt{T \cdot H(f^*)}$$

This demonstrates that the cumulative regret grows at a rate of $\mathcal{O}(\sqrt{T})$, which is sublinear in $T$. This result implies that while the total regret increases with step, the average regret per time step decreases at a rate of $\mathcal{O}(\frac{1}{\sqrt{T}})$.

# G ALGORITHMIC REALIZATION OF THE MIN-MAX FRAMEWORK

## G.1 PRACTICAL PROXIES FOR REGRET AND INFORMATION GAIN

Algorithm 1 provides a computationally tractable implementation of the abstract Min-Max optimization strategy presented in Eq. 1. It translates the theoretical concepts of regret minimization (exploitation) and information gain maximization (exploration) into concrete, efficiently computable metrics, enabling `PiFlow` to guide the scientific discovery process effectively. The bridge between theory and practice is established as follows:

**Approximating exploitation (minimizing cumulative regret).** The theoretical objective of minimizing cumulative regret, $\sum_{t=1}^{T}(v^* - f^*(h_t))$, is centered on favoring principles that consistently yield high-value outcomes $f^*(h)$. In our practical implementation, the `compute_exploitation_scores` function directly approximates this goal. It leverages the historical trajectory of principle-outcome pairs, $\mathcal{T}_t = \{\langle p_k, y_k \rangle\}_{k=1}^{t}$. The observed outcome $y_k$ here serves as a direct proxy for the performance of its corresponding principle $p_k$. A principle associated with higher outcomes is considered to have lower regret. Therefore, the resulting normalized score vector ranging from 0 to 1, $S_{exploitation} \in \mathbb{R}^{|\mathcal{T}_t|}$, quantifies the empirical success of each principle, and maximizing this score is equivalent to minimizing the cumulative regret based on past evidence.

**Approximating exploration (maximizing information gain).** The second term in our objective, maximizing the mutual information $I(h_t; f^*|H_{t-1})$, encourages the selection of hypotheses that are most informative about the underlying scientific landscape. A direct computation of mutual information is often intractable. Consequently, Algorithm 1 employs a practical proxy via the `compute_exploration_scores` function. **This is achieved by computing the semantic distance of principles using their sentence level embeddings (e.g., obtained from QwenMax model).** We posit that principles which are semantically distant from those already tested are more likely to reveal novel information about the function $f^*$. Therefore, we use the cosine distance between a candidate principle's embedding and the embeddings of previously explored principles as a heuristic for information gain. High exploration scores, $S_{exploration} \in \mathbb{R}^{|\mathcal{T}_t|}$, are assigned to conceptually novel principles. This heuristic is theoretically grounded, as detailed in Appendix G.2.

---

**Algorithm 1** Algorithm of `PiFlow`.

1: **Input:** $\mathcal{T}_t$, and $\lambda_{factor}$
2: **Output:** $suggestion$ (strategic action recommendation)
3: **if** $|\mathcal{T}_t| < 3$ **then**
4:      $suggestion \leftarrow$ "Initialize one principle to explore."
5: **else**
6:      $S_{\text{exploration}} \leftarrow$ compute_exploration_scores($\mathcal{T}_t$)
7:      $S_{\text{exploitation}} \leftarrow$ compute_exploitation_scores($\mathcal{T}_t$)
8:      **for** $i \leftarrow 1$ **to** $|\mathcal{T}_t|$ **do**
9:          $S_{\text{final}}[i] \leftarrow (1 - \lambda_{factor}) \cdot S_{\text{exploration}}[i] + \lambda_{factor} \cdot S_{\text{exploitation}}[i]$
10:      **end for**
11:      $i_{\text{best}} \leftarrow \arg\max_i(S_{\text{final}}[i])$
12:      $p_{\text{best}} \leftarrow \mathcal{T}[i_{\text{best}}]$
13:      $best\_exploitation\_score \leftarrow S_{\text{exploitation}}[i_{\text{best}}]$
14:      **if** $best\_exploitation\_score > 0.7$ **then**
15:          $action\_type \leftarrow$ "refine"
16:          $suggestion \leftarrow$ "Focus on refining: $\frown p_{\text{best}}.content$
17:      **else if** $best\_exploitation\_score > 0.4$ **then**
18:          $action\_type \leftarrow$ "validate"
19:          $suggestion \leftarrow$ "Validate: $\frown p_{\text{best}}.content$
20:      **else**
21:          $action\_type \leftarrow$ "explore"
22:          $suggestion \leftarrow$ "Explore alternatives: $\frown p_{\text{best}}.content$
23:      **end if**
24: **end if**
25: **return** $suggestion$

---

**The integrated decision policy.** The policy of the Min-Max framework, $\pi$, must balance the above objectives. Our algorithm materializes this policy through a two-step process:

1. **Step 1 (Scoring and selection).** The final score, $S_{final}[i] \leftarrow (1-\lambda_{factor}) \cdot S_{exploration}[i] + \lambda_{factor} \cdot S_{exploitation}[i]$, is a direct implementation of the balanced objective function. The input parameter $\lambda_{factor}$ instantiates the theoretical trade-off, controlling the emphasis between exploration and exploitation. The $\arg\max_i(S_{final}[i])$ operation then executes the policy's primary function: selecting the most promising principle, $p_{best}$, given the current state of knowledge and the desired strategic balance.

2. **Step 2 (Action recommendation).** Finally, the policy translates its choice into a strategic command. The threshold-based conditions on the best principle's exploitation score (if $best\_exploitation\_score > 0.7 \ldots$) discretize the continuous space of potential actions into three clear directives: **refine**, **validate**, or **explore** with the concatenation (denoted by $\frown$) of the principle content. This transforms the numerical output of the optimization into an actionable suggestion for the Planner agent ($\mathcal{A}_P$), thereby closing the loop between theoretical deliberation and practical execution within the Hypothesis-Validation cycle.

**Historical information management.** We manage historical information via two independent parts: (a) agent memory for MAS to iterate the hypothesis-testing and (b) persistent principle pool for PiFlow layer to guide the process.

1. **For MAS part, agent history differs per agent.** As the MAS follows round-robin order, there is the risk of context collapse. To prevent this, each agent maintains an independent MessageBuffer, retaining the most recent $M = 10$ messages. This ensures agents focus on the immediate logical flow without being overwhelmed by noise.

2. **For `PiFlow`, as a plugin layer, it has its own memory.** Crucially, our principle-aware design decouples reasoning history from scientific insight. While chat logs may be truncated, generated principles and validation scores represent compressed, high-value knowledge stored in a separate, persistent global pool. As per Algorithm 1 (Lines 14-23), even if an agent forgets a specific past conversation, PiFlow retrieves the refined principles from the global pool to guide the next step. This ensures the logic chain remains intact throughout the discovery process.

With the design of decoupling, PiFlow only need to collect historical principles, hypotheses and outcomes to optimize the strategy (explore, refine or validate which temporary principle), and steers the learning process by giving guidance to MAS.

**Realworld efficiency (stopping by condition).** In practical deployment, PiFlow's Min-Max optimization naturally drives the system toward convergence, enabling precise threshold-based stopping: In the Molecular Bio-activity Optimization (MBO) task, PiFlow-MAS improved pChEMBL from 2.69 to 6.44 within just 4 effective iterations, and surpassed the target (6.5) to reach 7.65 by the 6th iteration. This demonstrates that with a practical stop condition, PiFlow achieves success using only 25% of the allocated benchmark budget. As detailed in Section 5.4, our Min-Max framework provides a theoretical guarantee that prevents infinite loops of low-quality exploration, ensuring resource efficiency even in the intensive tasks noted by the Reviewer.

## G.2 THE RATIONAL OF SEMANTIC DISTANCE AS A PROXY FOR INFORMATION GAIN

Recall that we introduce a Min-Max optimization framework for the `PiFlow` system. The objective function, as shown in Eq. 1, incorporates a mutual information term, $I(h_t; f^*|H_{t-1})$, to guide exploration. This term, while theoretically ideal, is computationally intractable as it requires knowledge of the true underlying evaluation function $f^*$. In our practical implementation, we employ a surrogate objective for exploration: maximizing the distance of a new principle's text embedding from the embeddings of previously selected principles. Here, we provide a formal justification for this approximation, demonstrating its theoretical soundness.

*Proof.* Our goal is to establish a principled connection between the practical exploration strategy and the theoretical objective of maximizing information gain. The practical strategy is to select a principle $p_t$ from the principle space $\mathcal{P}$ at each timestep $t$ according to:

$$p_t^* = \arg\max_{p_t \in \mathcal{P}} \left( \min_{m \in \{1, \ldots, t-1\}} \|\phi(p_t) - \phi(p_m)\|_2 \right) \tag{4}$$

where $\phi : \mathcal{P} \to \mathbb{R}^d$ is a function that maps a principle to its corresponding high-dimensional text embedding. We will now demonstrate that this objective serves as a valid proxy for maximizing the mutual information term $I(h_t; f^* | H_{t-1})$.

The derivation rests upon the hierarchical relationship between principles and hypotheses, and the semantic properties of modern language model embeddings. A principle $p \in \mathcal{P}$ is not a hypothesis itself, but rather defines a specific semantic region or a conditional distribution $\pi(\cdot | p)$ from which a concrete hypothesis $h \in \mathcal{H}$ is formulated. Thus, the selection of a principle $p_t$ precedes the generation of a hypothesis $h_t \sim \pi(\cdot | p_t)$.

We begin by positing two fundamental premises regarding the nature of the embedding space.

**Premise 1 (Semantic-metric correspondence).** The embedding function $\phi$ is assumed to map the conceptual space of principles to a metric space where distance reflects semantic dissimilarity. That is, for any two principles $p_i, p_j \in \mathcal{P}$, a large Euclidean distance $\|\phi(p_i) - \phi(p_j)\|_2$ implies a significant divergence in their underlying semantic and conceptual content. This is a well-established property of embeddings from large-scale language models.

**Premise 2 (Functional consequence of semantic dissimilarity).** Semantically distinct principles guide the generation of functionally distinct hypotheses. A hypothesis $h$ acts as a probe of the unknown function $f^*$. If two principles $p_i$ and $p_j$ are semantically distant, the hypotheses generated from their respective distributions, $h_i \sim \pi(\cdot | p_i)$ and $h_j \sim \pi(\cdot | p_j)$, are expected to probe disparate aspects or regions of the function $f^*$.

With these premises, we can construct the logical argument. The mutual information term $I(h_t; f^* | H_{t-1})$ quantifies the expected reduction in uncertainty about $f^*$ after observing the outcome of hypothesis $h_t$, given the history of observations $H_{t-1}$. As established in our convergence proof (Section F), this information gain is related to the posterior variance of the outcome of $h_t$:

$$\mathbb{E}\left[ (v^* - f^*(h_t))^2 \mid H_{t-1} \right] \leq c \cdot I(h_t; f^* \mid H_{t-1})$$

This relation suggests that **a hypothesis $h_t$ yielding high uncertainty (i.e., high posterior variance $\mathbf{Var}(f^*(h_t) \mid H_{t-1})$) is expected to provide high information gain. Consequently, a sound exploration strategy is to select $h_t$ to maximize this variance.**

Let us now connect this objective to our practical strategy in Eq. 4.

1. By selecting a principle $p_t$ that maximizes the minimum distance to all prior principles in the embedding space, we are, by Premise 1, choosing a principle that is maximally semantically novel compared to the history of principles $\{p_m\}_{m=1}^{t-1}$.

2. By Premise 2, this semantically novel principle $p_t$ will guide the generation of a hypothesis $h_t$ that probes a functionally distinct aspect of $f^*$ compared to all prior hypotheses in $H_{t-1} = \{(h_m, y_m)\}_{m=1}^{t-1}$.

3. Since $h_t$ lies in a region of the hypothesis space that has not been explored by past observations, our model's posterior belief about the outcome $f^*(h_t)$ will be characterized by high uncertainty. This high uncertainty mathematically corresponds to a large posterior variance, $\mathrm{Var}(f^*(h_t) \mid H_{t-1})$.

4. Therefore, the policy of maximizing the embedding distance effectively drives the selection of hypotheses that are expected to have high posterior variance.

This chain of above analysis leads to the following correspondence:

$$\arg\max_{p_t} \left( \min_{m<t} \|\phi(p_t) - \phi(p_m)\| \right) \Rightarrow \text{Select maximally novel } p_t$$

$$\Rightarrow \text{Generate } h_t \text{ from an unexplored hypothesis subspace}$$
$$\Rightarrow \max_{h_t \sim \pi(\cdot|p_t)} \mathbb{E}[\text{Var}(f^*(h_t) \mid H_{t-1})]$$
$$\Rightarrow \max_{h_t \sim \pi(\cdot|p_t)} \mathbb{E}[I(h_t; f^* \mid H_{t-1})]$$

The expectation $\mathbb{E}[\cdot]$ is taken over the generation of $h_t$ from $p_t$.

In conclusion, the strategy of maximizing the minimum embedding distance between principles is not an arbitrary heuristic. It is a principled and computationally feasible surrogate for the theoretically-grounded objective of maximizing mutual information. It leverages the semantic structure captured by modern language model embeddings to implement an efficient and effective exploration strategy, ensuring that `PiFlow` diversifies its inquiry at a conceptual level. This alignment between our practical approximation and the theoretical Min-Max framework provides support for the design of our system and its empirical performance. □

> **Take-away**: We operationalize the abstract Min-Max framework by substituting its theoretically-ideal but computationally-intractable objectives with practical proxies. **Exploitation** (regret minimization) is approximated by historical performance, while **Exploration** (information gain maximization) is driven by maximizing the semantic distance between principle embeddings. We formally prove that this semantic distance is a principled surrogate for maximizing information gain, thus bridging the gap between theory and efficient implementation.

## H  ILLUSTRATIVE EXAMPLE: APPLICATION TO NANOHELIX OPTIMIZATION

We provide a step-by-step walk-through of `PiFlow`'s operation using the Nanohelix Optimization (NHO) task as a running example:

Let's assume the goal is to find the nanohelix geometry (described at Appendix S.1) with the maximum g-factor:

**Phase 1: Initial Exploration.**   The process begins with an unguided hypothesis generation to build an initial evidence base by LLM itself. The Planner agent, following the initial directive of `PiFlow`, instructs the Hypothesis Agent for intuitive exploring (Algorithm 1, line 3-5). For the first $K$ rounds (e.g., $K = 3$ in our experiments), the loop proceeds as follows,

1. **Hypothesis Agent** proposes a specific, testable hypothesis, *"Based on the principle of $\cdots$, measure the g-factor of a nanohelix with parameters: fiber_radius=40.0 nm, helix_radius=70.0 nm, n_turns=6.5, pitch=130.0 nm"*.

2. **Experiment Agent** calls the tool (surrogate model) to validate the above hypothesis and returns the outcome, *"the validation yields a g-factor of 0.86"*.

3. This Hypothesis-Validation process repeats for $K$ rounds, populating the evidence trajectory $T_t$ for establishing an initial principle pool.

In the initial $K$ rounds, lacking specific guiding principles, the Hypothesis Agent performs an initial exploration of the parameter space by proposing diverse geometries based on its general pre-trained knowledge, or by sampling from a wide distribution.

**Phase 2:  Principle-aware guidance from PiFlow.**   After the initial $K$ rounds, the MinMax optimization of `PiFlow` parses these evidence and identifies the highest potential principle and suggests an action based on the score of the highest-potential principle (refer to Algorithm 1, line 5-23). The Planner agent receives this strategic guidance, for instance: "***Refine** the identified promising principle: By adjusting the fiber radius to twice the helix radius and fine-tuning other parameters, it is expected to maximize the chirality (g factor) of the nanohelix*".

**Structured, guided hypothesizing in both Phase 1 and Phase 2.**   The Planner injects this directive into the whole group chatting history, i.e., Hypothesizing-Validation loop, thereby reducing the system-level uncertainty. The Hypothesis Agent now uses this focused principle to formulate its next hypothesis with structured reasoning, for example:

1. **Major premise.** The g-factor of a nanohelix is governed by the spatial asymmetry and electromagnetic coupling arising from its geometric parameters.

2. **Minor premise 1.** The previously tested geometry (fiber-radius=40 nm, helix-radius=70 nm, n-turns=6.5, pitch=130 nm) yielded a high g-factor ( 0.86), indicating a favorable parameter balance.

3. **Minor premise 2 (Inspired by PiFlow).** Systematically varying parameters around this near-optimal point can induce nonlinear electromagnetic effects, potentially increasing chirality. Specifically, increasing the pitch and number of turns might enhance the chiral interaction length.

4. **Proposed testable hypothesis.** "Measure the g-factor of a nanohelix with parameters: fiber_radius=49.0 nm, helix_radius=24.5.0 nm, n_turns=7.0, pitch=140.0 nm."

By this way, `PiFlow` dynamically steers the discovery process away from aimless searching and towards a focused, principle-driven exploration, thereby systematically accumulating information and avoiding inefficient exploration directions.

---

**Takeaway**: In this example, `PiFlow` transforms the optimization process from an initial, unguided exploration into a focused, principle-driven discovery. It first samples the parameter space to build an empirical evidence base, then distills a high-potential guiding principle to steer subsequent hypotheses, ensuring a systematic and efficient search for the optimal nanohelix geometry.

---

# I ANALYSIS OF BASELINES RESPONSE

## I.1 RESPONSE ANALYSIS OF BASELINES

We compare our PiFlow against the ReAct, MPO and Vanilla Agent. Through the experiment, we found that the MPO baseline incorporates a form of global, step-by-step reasoning or reflection within their trained model, for example, it's exact output is, for example, *"Step 1: Identify the task objective. Step 2: Survey the environment. Step 3: Consider the first potential candidate. Step 4: Shift to the next potential candidate. Step 5: Repeat Steps 3 and 4 until the task objective is met. Step 6: Confirm the task completion."*

While ReAct responses with the reflection of "Previous Experiment Results" and then instructs the Hypothesis Agent to try another new hypothesis directly with the provided "Rationale", the baseline of Vanilla Agent, in genral, response with step-by-step thoughts following "Step 1: Define the Hypothesis, Step 2: Initial Exploration, Step 3: Parameter Space Definition, Step 4: Experimental Design, Step 4: Exact Experiments (candidates)", behaving like a careful-thought planner in the whole process of scientific discovery.

Backend these outputs, the performance difference stems from the level of dynamic and strategic guidance. For example, **the reasoning of MPO and two AI Scientsit methods are local** and **tactical**. MPO generates a fixed "how-to" checklist. This plan dictates the operational steps but is agnostic to the underlying scientific principles driving the experiments, and cannot reason about why its plan is failing, leading to hypotheses that yield lower outcomes. While The-AI-Scientist-v2 and AI-Researcher logically select the "node" or candidate, their lack of hypothesis diversity often leads to **premature convergence and performance stagnation**. Additionally, the Vanilla agent uses the exact same powerful LLM (QwenMax) and has access to the same experimental tools. Its poor performance demonstrates that **even a capable LLM, without strategic guidance, explores aimlessly**. The dramatic performance gap between Vanilla and PiFlow (e.g., AUC improving from 35.96% to 63.51% in the NHO task) is direct evidence that our contributions, i.e., (a) structuring knowledge into an explicit principle space, and (b) applying a rigorous Min-Max optimization strategy to navigate it.

However, `PiFlow` adapts its learnable strategy. By optimizing (i.e., evaluating and selecting) at the principle level, `PiFlow` can make strategic jumps. When hypotheses derived from Principle A consistently fail, the Min-Max optimization quantitatively lowers the score of Principle A itself, guiding the agent to switch to a completely different research direction (Principle B). This prevents getting stuck and is the fundamental reason for its superior efficiency and performance, as demonstrated empirically in Figure 3.

## I.2 Statistical Significant Test on Metrics

We compared PiFlow against AI-Researcher (Tang et al., 2025) (NeurIPS 2025) and The-AI-Scientist-v2 (Yamada et al., 2025). As shown in Table 5, `PiFlow`'s targeted uncertainty reduction (Min-Max) proves superior to generic autonomous loops.

Table 5: Performance comparison between `PiFlow`, The-AI-Scientist-v2, and AI-Researcher.

| Task | Method | SQ (%) | P-value (SQ) | AUC (%) | P-value (AUC) |
|------|--------|--------|--------------|---------|---------------|
| **NHO** | PiFlow | **76.82 ± 4.54** | – | **63.51 ± 11.18** | – |
| | The-AI-Scientist-v2 | 56.67 ± 5.79 | 0.0103 | 49.27 ± 1.84 | 0.1547 |
| | AI-Researcher | 53.12 ± 1.06 | 0.0091 | 46.45 ± 2.42 | 0.1122 |
| **MBO** | PiFlow | 84.55 ± 29.63 | – | **64.57 ± 23.65** | – |
| | The-AI-Scientist-v2 | 62.47 ± 8.02 | 0.3250 | 36.32 ± 10.62 | 0.1630 |
| | AI-Researcher | **95.66 ± 7.45** | 0.5870 | 15.56 ± 12.71 | 0.2625 |
| **SPO** | PiFlow | **34.85 ± 1.19** | – | 21.51 ± 2.80 | – |
| | The-AI-Scientist-v2 | 29.85 ± 2.68 | 0.0663 | **28.43 ± 3.13** | 0.0470 |
| | AI-Researcher | 25.69 ± 3.23 | 0.0277 | 16.36 ± 2.72 | 0.0842 |

The results show that, **`PiFlow` achieves statistically significant improvement (p-value $< 0.05$) on NHO and SPO tasks.** On the MBO task, PiFlow remains competitive with higher efficiency (AUC 64.57% vs 42.72%), though SQ variance is higher. This confirms the advantage of `PiFlow` in complex, uncertainty-heavy environments.

> **Take-away**: Baselines fail due to their reliance on rigid, tactical plans, which leads to aimless exploration when a strategy is ineffective. In contrast, `PiFlow` excels by operating at a higher level of abstraction. It reasons over a "*principle space*" and employs Min-Max optimization to strategically pivot away from failing research directions, enabling adaptive and efficient discovery.

## J  Performance Comparison with Bayesian Optimization

To validate our hypothesis that a principle-aware architecture is superior for scientific discovery under high epistemic uncertainty, we compared `PiFlow` against a strong, general-purpose baseline, Bayesian Optimization (BO). It is worth noting that, unlike `PiFlow`, BO requires significant expert effort to manually define a parameterized optimization space. We construct the experiment by manually configuring the searching space of BO in both nanohelix optimization (NHO) task, molecular bio-activity optimization task and superconductor optimization (SPO) task.

Table 6: Performance comparison between `PiFlow` and BO (mean ± standard deviation)

| Method | Metric | NHO | MBO | SPO |
|--------|--------|-----|-----|-----|
| PiFlow | AUC | 63.51 ± 11.18 | 46.11 ± 16.25 | 21.51 ± 2.80 |
| | SQ | 76.82 ± 4.54 | 84.55 ± 29.63 | 34.85 ± 1.19 |
| BO | AUC | 68.86 ± 5.86 | 34.76 ± 4.21 | 31.61 ± 0.19 |
| | SQ | 78.76 ± 0.77 | 38.15 ± 4.02 | 32.38 ± 0.32 |

Results with 24 iterations of BO, summarized in Table 1, show that while BO is competitive on the structured NHO task, it strongly needs prior design of the searching space, e.g., material element composition and quantity range. However, `PiFlow` demonstrates a substantial performance advantage on the more complex and uncertain MBO task, highlighting its effectiveness when the problem structure is not known a priori.

In fact, `PiFlow` leverages scientific principles with its dynamic uncertainty reduction architecture. This framework allows the agent to make "domain-shifting" leaps in the search space (e.g., the jump

from a 60nm to a 10nm radius value), leading to rapid, significant gains that are rewarded by the AUC metric. It is this ability to build and reason with an evolving knowledge structure that our evaluation framework was designed to highlight.

In summary, `PiFlow` addresses the challenge of semantic hypothesis generation, where BO, Generic Algorithms and Reinforcement Learning methods face significant barriers in representation. PiFlow operates in an open-ended semantic space (natural language principles), allowing it to be Plug-and-Play without domain-specific encoding.

---

**Take-away**: `PiFlow`'s principle-aware architecture significantly outperforms Bayesian Optimization on complex, ill-structured scientific discovery tasks (MBO). Its strength lies in dynamically building knowledge to navigate vast search spaces, whereas BO's performance is contingent on a manually-defined search space, rendering it only competitive on well-structured problems (NHO).

---

## K  PLUG-AND-PLAY INTEGRATION WITH CHEMTOOLAGENT

We integrate `PiFlow` with ChemToolAgent (Yu et al., 2024) on the Molecular Bio-activity Optimization (MBO) task to demonstrate the generalization of `PiFlow`.

The integration follows a Plug-and-Play design, where PiFlow acts as a strategic guidance layer, as shown in Figure 5. ChemToolAgent serves as both the **Hypothesizer** and Experimenter, leveraging its built-in tools (such as searching molecules' formula from PubMed and website) to propose molecular hypotheses, while `PiFlow`'s outer-loop guidance provided EXPLORE and REFINE commands to ensure systematic and efficient discovery dynamically. As shown in Table 7, this collaborative process successfully evolved its strategy from basic principles to a high-value molecular design (pChEMBL of 5.90) over 8 iterations **without any architecture-level modifications** over ChemToolAgent, demonstrating a clear synergy.

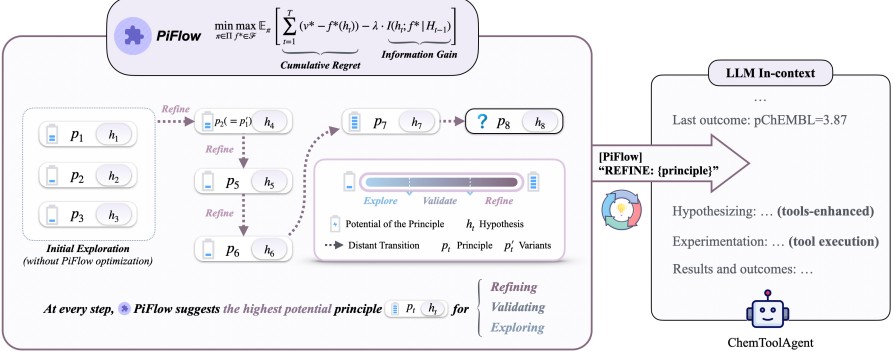

Figure 5: Illustration of combining `PiFlow` with existing MAS (ChemToolAgent) through Plug-and-Play.

---

**Take-away**: `PiFlow` seamlessly integrates with existing agents like ChemToolAgent as a zero-modification, plug-and-play strategic layer. By providing high-level EXPLORE and REFINE guidance, it synergistically steers the discovery process from general principles to a high-potency molecule (pChEMBL 5.90), demonstrating its strong generalization and practical value.

---

## L  THEORETICAL COMPUTATIONAL COMPLEXITY

Let $t$ be the current number of iterations (i.e., the size of the historical trajectory) and $d$ be the embedding dimension of the principles. A single decision step in `PiFlow` (Algorithm 1) involves:

1. Computing exploitation scores. This requires iterating through the t historical outcomes, giving a complexity of $\mathcal{O}(t)$.

Table 7: Key steps from the 8-iteration integration run with ChemToolAgent.

| Iteration | PiFlow Action | Principle / Key Guidance | pChEMBL |
|---|---|---|---|
| 1 | EXPLORE | Proposed $p_1$: *"hydroxyl + nitrogen heterocycle for H-bonding"* | 2.80 |
| 2 | EXPLORE | Tested a new, unrelated principle $p_2$ | -0.10 |
| 3 | EXPLORE | ChemToolAgent generated a principle $p_3$ and proposed an invalid formula | null |
| 4 | REFINE on $p_1$ | Focused on the promising principle ($p_1$) from the Iteration 1 | 2.13 |
| 5 | REFINE on $p_1$ | Proposed $p_5$: *"Quinazoline + phenyl rings for hydrophobic interactions..."* | 3.87 |
| 6 | REFINE on $p_5$ | Further refined the quinazoline principle (dual EGFR/HER2 mechanism) | 5.90 |
| 7-8 | REFINE | Continued focused refinement on the high-potential quinazoline scaffold | ~5.2 |

2. Computing exploration scores. To assess the novelty of each of the $t$ historical principles, the default algorithm calculates its similarity to all other $t-1$ principles. This involves $\mathcal{O}(t^2)$ vector comparisons, leading to a complexity of $\mathcal{O}(t^2 \cdot d)$.

3. Final decision. This involves a weighted sum and finding the maximum over t principles, costing $\mathcal{O}(t)$.

Therefore, the dominant computational cost for a single PiFlow decision at step $t$ is $\mathcal{O}(t^2 \cdot d)$. While manageable for moderate trajectory $\mathcal{T}$, we recognize this can be a bottleneck for very long process.

To ensure scalability, this can be readily optimized to near-linear complexity using standard techniques, such as (a) instead of all-pairs comparison, we can find the most similar principles for approximation, aiming for reducing the exploration score computation; (b) incremental or batched updates of similarity matrix, avoiding full recalculation at every step. These optimizations make `PiFlow` computationally feasible for large-scale discovery tasks.

**The decision complexity of $O(t^2 \cdot d)$ is not a bottleneck.** We profiled the runtime in the NHO task. Over 24 iterations, the entire `PiFlow` framework accounted for only 9.4% of the total PiFlow-MAS runtime, while the surrogate model took 0.2%. We also quantitatively profiled the optimization efficiency in the NHO task. The result confirms that the algorithmic complexity of `PiFlow` translates into superior search efficiency: For baselines, Vanilla Agent System achieves SQ=50% taking 2416s; The other baseline MPO achieves SQ=52% taking 506s and baseline ReAct even fails to reach such level of SQ in all its 918s. However, our **`PiFlow` achieves SQ=52% (similar bar) by only 427s, delivering such a result using less than 1/5 of the Vanilla Agent's time (a 5.6x speedup), and is 1.2x faster than the baseline MPO.**

The results indicate that, the vast majority of time is consumed by LLM inference. The specific similarity calculation ($t^2$) takes milliseconds, which is structurally distinct from—and negligible compared to—LLM generation (seconds) or real-world wet-lab experiments (hours/days).

> **Take-away**: The per-step complexity of `PiFlow` is $\mathcal{O}(t^2 \cdot d)$, dominated by an all-pairs similarity calculation for exploration. This quadratic cost is not a fundamental limitation, as it can be readily optimized to near-linear time, ensuring scalability.

## M  COST-EFFECTIVENESS ANALYSIS

We performed a cost-effectiveness analysis comparing our **PiFlow-MAS** against a **Vanilla-Agent System** baseline across three discovery tasks (NHO, MBO, SPO). All experiments were run for 24 iterations using the Qwen-Max model. Table 8 compares the total token consumption (cost) and final

Solution Quality (SQ) achieved, while Table 9 isolates the token cost of the PiFlow module itself to demonstrate its efficiency. Both of them are shown at Figure 6.

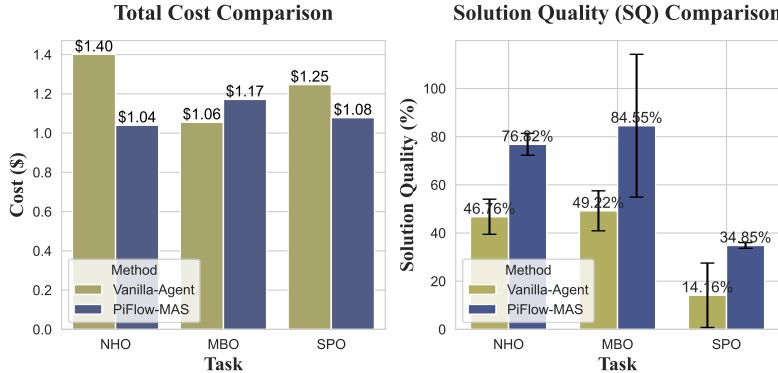

Figure 6: Cost-effectiveness and solution quality (SQ) comparision.

Table 8: Cost-effectiveness comparison between PiFlow-MAS and a Vanilla-Agent System baseline.

| Task | Method | Tokens | Cost ($) | Cost/Iter ($) | Cost Reduction (%) | SQ (%) |
|------|--------|--------|----------|---------------|---------------------|--------|
| NHO | Vanilla-Agent | 806,949 | $1.4011 | $0.0584 | - | $46.76 \pm 7.29$ |
|     | PiFlow-MAS | 591,316 | $1.0400 | $0.0434 | 26.7% | $76.82 \pm 4.54$ |
| MBO | Vanilla-Agent | 610,208 | $1.0552 | $0.0440 | - | $49.22 \pm 8.30$ |
|     | PiFlow-MAS | 657,594 | $1.1717 | $0.0493 | -7.7% | $84.55 \pm 29.63$ |
| SPO | Vanilla-Agent | 707,829 | $1.2469 | $0.0520 | - | $14.16 \pm 13.37$ |
|     | PiFlow-MAS | 610,284 | $1.0785 | $0.0449 | 13.7% | $34.85 \pm 1.19$ |

Table 9: Analysis of the PiFlow module's token efficiency as a lightweight plugin.

| Task | PiFlow-MAS | PiFlow | PiFlow Token Share (%) | PiFlow Tokens/Iter |
|------|-----------|--------|------------------------|---------------------|
| NHO | 582,396 | 8,920 | 1.5% | 372 |
| MBO | 649,590 | 8,004 | 1.2% | 334 |
| SPO | 602,740 | 7,544 | 1.2% | 314 |

**Takeaway: PiFlow-MAS achieves superior performance at a reduced cost.** Our analysis reveals that PiFlow-MAS significantly enhances Solution Quality (SQ) by up to **2.4x** while simultaneously cutting token consumption by up to **26.7%**. This is accomplished with remarkable efficiency, as the PiFlow module itself is a lightweight plugin, constituting only **1.2-1.5%** of the total tokens.

## N ABLATION: TEMPORAL DYNAMICS OF PRINCIPLE EVALUATION

We provide an empirical visualization of the internal dynamics of `PiFlow`, connecting directly to the Min-Max optimization framework detailed in Section 3.3. To illustrate how the scores for different scientific principles evolve over time, we present results from an implementation using the QwenMax model on the NHO (nanohelix optimization) benchmark. The following figures chart the iterative recalculation of principle scores, which guides the system's strategic balance between exploitation and exploration.

Figure 7 displays the dynamic final scores ($S_{final}$) for each principle, representing the overall potential as determined by the Min-Max optimization with $\lambda = 0.5$. These scores are not static; they fluctuate as the Hypothesis-Validation loop accumulates new evidence. **Some principles that initially appear promising see their scores decline, while others emerge as high-potential candidates over**

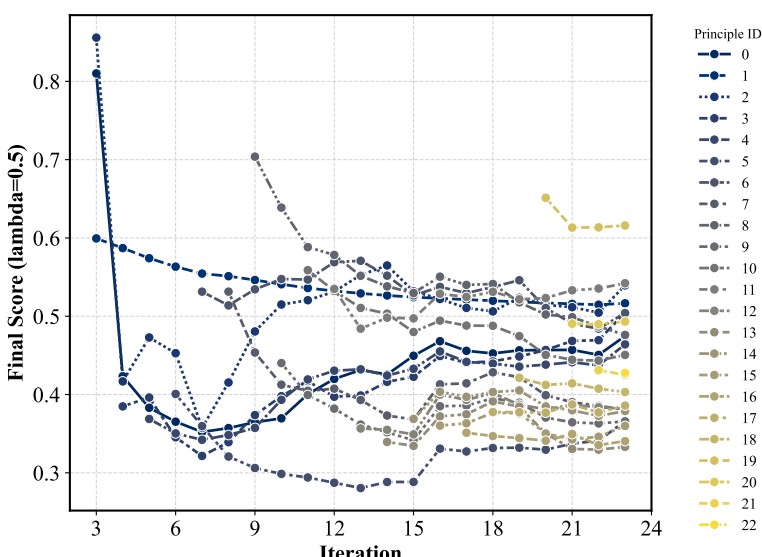

Figure 7: The dynamic final scores of principles in `PiFlow`.

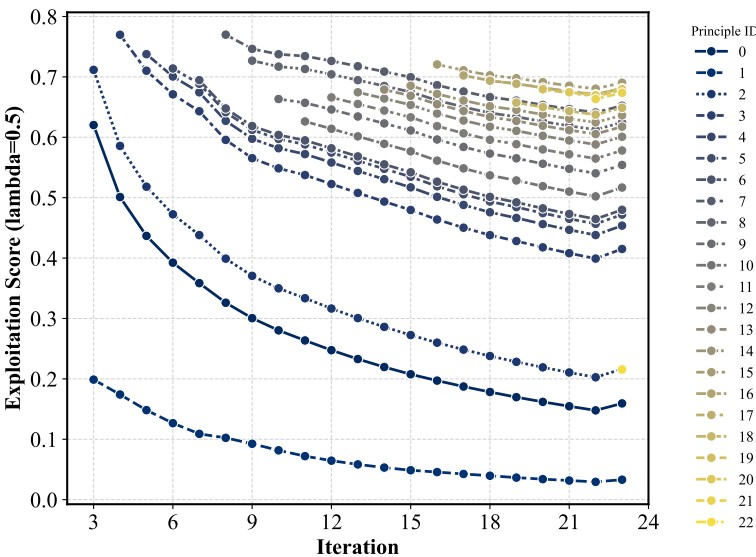

Figure 8: The dynamic exploitation scores of principles in `PiFlow`.

**subsequent iterations.** This dynamic ranking is the direct output of `PiFlow`'s strategic analysis, steering the discovery process toward the most promising avenues at any given moment.

To better understand the final scores, Figures 8 and 9 decompose them into their constituent exploitation and exploration components, respectively.

Specifically, Figure 8 shows that the exploitation scores for most principles tend to decrease over time. This reflects the regret-minimization objective; as principles are tested and accumulated to evidence (exploited), the potential for further high-value discoveries from these experience may diminish, or the cumulative regret associated with it increases. This manifests itself **as principles generated later in the hypothesis testing process naturally have higher outcomes**, while past principles are reduced due to normalization.

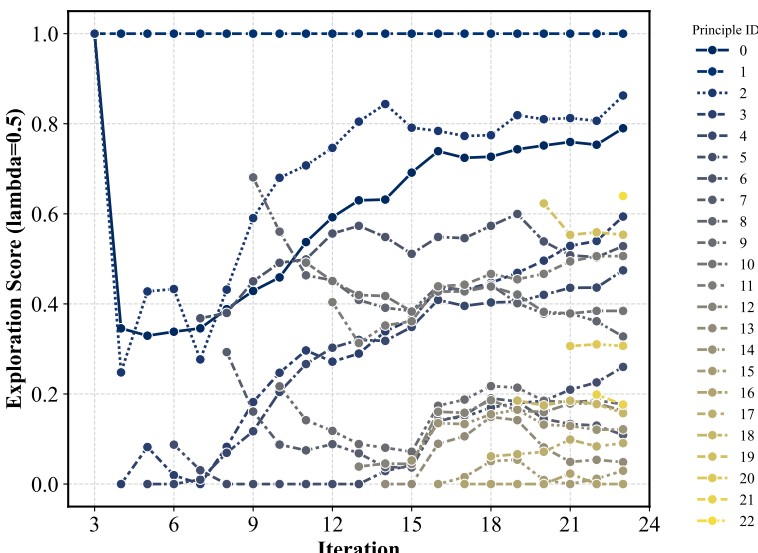

Figure 9: The dynamic exploration scores of principles in `PiFlow`.

In contrast, Figure 9 reveals a much more varied and dynamic behavior for the exploration scores. Principles may begin with a low exploration score and later see it increase significantly, indicating that the system has identified a knowledge gap and is prioritizing information gain to reduce uncertainty about that principle's potential.

Together, these plots illustrate the practical outcome of the adversarial optimization: a continuous, adaptive balancing act where `PiFlow` shifts its focus between exploiting known concepts and exploring uncertain ones to navigate the scientific landscape efficiently.

> **Takeaway:** Principle evaluation in `PiFlow` is a dynamic balancing act. The system continuously re-calibrates the trade-off between exploiting known concepts (diminishing returns) and exploring uncertain ones (information gain) to adaptively steer the discovery process.

## O   ABLATION: THE IMPACT OF FOUNDATION MODELS

**Influence of Foundation Models.**   To evaluate the impact of different models, we replace LLMs of $\mathcal{A}_H$ and $\mathcal{A}_P$ evaluation. The $\mathcal{A}_E$ is responsible for tool usage, consistently used QwenMax to ensure functional tool interaction. We evaluate several state-of-the-art LLMs (e.g., Claude-3.7-sonnet (Anthropic, 2025), GPT4.1-mini (OpenAI, 2025), Gemini-2.5-pro-exp-0325 (DeepMind, 2025), Qwen3-32B and QwenMax (Yang et al., 2024b)) on the Nanohelix Optimization (NHO) task with three times of random seeds as initialization, and the results are summarized in Table 10.

Table 10: Ablation Study of Model Types (mean $\pm$ std)

| Method/Setting | AUC (%) | SQ (%) |
|---|---|---|
| Claude-3.7-sonnet | 38.60 $_{\pm 4.30}$ | 78.50 $_{\pm 3.74}$ |
| GPT4.1-mini | 41.68 $_{\pm 17.91}$ | 66.38 $_{\pm 14.90}$ |
| Gemini-2.5-pro-exp-03-25 | 28.43 $_{\pm 13.81}$ | 69.64 $_{\pm 17.10}$ |
| Qwen3-32B | 37.51 $_{\pm 7.70}$ | 58.76 $_{\pm 6.18}$ |
| QwenMax | **63.51** $_{\pm 11.18}$ | **76.82** $_{\pm 4.54}$ |

Among these models, QwenMax demonstrates the highest AUC at 63.51% and a strong SQ of 76.82%. Claude-3.7-sonnet achieves the highest SQ at 78.50% with an AUC of 38.60%. Other models like GPT4.1-mini (AUC 41.68%, SQ 66.38%) and Qwen3-32B (AUC 37.51%, SQ 58.76%)

also show competent performance. Gemini-2.5-pro-exp-03-25 yields an AUC of 28.43% and an SQ of 69.64%. These quantitative results reveal that, while efficiency (AUC) is model-dependent—ranging from 38.60% (Claude-3.7) to 63.51% (QwenMax), Solution Quality (SQ) benefits consistently from stronger reasoners.Specifically, Claude-3.7-sonnet achieves the highest SQ of 78.50%, outperforming the efficiency-optimized QwenMax (76.82%) by $\sim 1.7\%$.

In summary, these results confirm that PiFlow generalizes well, effectively converting superior base model capabilities into higher-quality optimization outcomes.

> **Takeaway:** The choice of the foundation model is critical. Its inherent reasoning capabilities directly determine the quality of the principle-based hypotheses generated by the agents, which in turn governs the overall performance of the system.

## P  ABLATION: THE IMPACT OF INITIAL PRINCIPLE QUALITY

In this section, we provide a detailed analysis of the performance under two challenging scenarios to evaluate the robustness and the practical implications of the underlying exploration-exploitation mechanism in `PiFlow`. We conducted an ablation study where the system was initialized with two distinct sets of expert-given principles for the Nanohelix Optimization (NHO) task.

### P.1  EXPERIMENT SETTINGS

We use the QwenMax model with the same settings as reported in the main experiments, repeating each scenario three times with different random seeds. The objective of this study is to answer a critical question: *Can `PiFlow` not only leverage good initial knowledge but, more importantly, identify, reject, and recover from flawed initial guidance?* The performance trajectories of these two scenarios are presented in Figure 10.

**Human-given correct principles.**    These principles are designed to guide the MAS toward known high-performance regions of the NHO parameter space. We derived them from a preliminary analysis of the surrogate model and established physical intuitions about chiroptical phenomena, see Table 11 Expert-Correct. Specifically, they encode only correct parameter correlations, such as the positive correlation between g-factor and parameters like pitch and number of turns, and guide the search towards previously identified optimal regimes (about 90% of the $\mu_{\text{absolute}}^{g-factor}$) for fiber radius and helix radius. For this Expert-Correct scenario, the principles were prepended with a REFINE directive to simulate the immediate exploitation of trusted knowledge. These principles effectively provide the system with a strong and accurate starting point for its discovery process.

**Human-given incorrect principles.**    These principles were manually constructed to deliberately mislead the MAS into low-performance regions, see Table 11 Expert-Incorrect. Through preliminary experiments, we also identified incorrect parameter correlations that consistently led to hypotheses with outcomes in the bottom 10% of the $\mu_{\text{absolute}}^{g-factor}$. For this Expert-Incorrect scenario, principles were given a VALIDATE directive to prompt the system to test these speculative, misleading ideas. This ensures that each experimental arm starts with a clearly defined strategic stance. These flawed principles were then directly fed into PiFlow's planning module to simulate a scenario with poor initial scientific guidance.

### P.2  RESULTS

As illustrated in Figure 10, the two scenarios tell a compelling story about PiFlow's operational dynamics. We can dissect the process into three key phases:

**Initial phase (Iteration 0-7).**    During the initial phase, the system's behavior is heavily influenced by the provided principles, leading to drastically different starting performances:

    a. **Expert-Correct.** The trajectory begins at a very high solution quality ($\sim$80%), demonstrating the system's ability to effectively exploit high-quality knowledge for immediate gains.

Table 11: Initial expert-given principles for the robustness study

| Scenario | ID | Principle Statement |
|---|---|---|
| Expert-Correct | 1 | REFINE: The g-factor is strongly enhanced by maximizing the axial pitch and the number of turns, as this elongates the helical structure and increases the effective interaction length for circularly polarized light. |
| | 2 | REFINE: Optimal g-factor enhancement occurs in two distinct regimes of fiber radius, corresponding to the selective excitation of different plasmon resonance modes: a narrow-radius regime (SPP coupling) and a wide-radius regime (LSPR effects). |
| | 3 | REFINE: The g-factor is critically dependent on the helix radius, which governs the coupling strength between adjacent turns. A compact helix radius (e.g., 20 nm) appears optimal for at least one major resonant regime. |
| Expert-Incorrect | 1 | VALIDATE: The most stable structures are formed when a geometric harmony exists where the pitch is approximately twice the helix radius (Pitch $\approx$ 2 times Helix Radius). |
| | 2 | VALIDATE: To maintain optimal activity, there must be a trade-off between the fiber's thickness and its length (number of turns). Increasing the fiber radius necessitates a decrease in the number of turns, and vice versa. |
| | 3 | VALIDATE: Optimal mode coupling occurs when the geometry is self-similar. Therefore, the system should prioritize configurations where the helix radius and fiber radius are as close in value as possible (Helix Radius $\approx$ Fiber Radius). |

b. **Expert-Incorrect.** In contrast, the trajectory languishes at a very low performance level ($<20\%$). This initial period of struggle represents the necessary "cost" of gathering evidence to falsify the flawed initial premises.

**Transition phase (Iteration 7-14).**    This phase reveals the Min-Max optimization of `PiFlow`, prompting a strategic shift in both scenarios:

a. **Expert-Correct.**    Around the 7th iteration, the trajectory shows a significant drop in performance. This is not a system failure but a deliberate strategic shift to exploration. Having exhausted the immediate benefits of the initial principles, the framework compels the system to prioritize **long-term information gain over short-term rewards** to avoid premature convergence on a local optimum.

b. **Expert-Incorrect.**    Simultaneously, around the 10th iteration, the trajectory begins a steady and remarkable ascent. This marks the point where the system has accumulated sufficient contradictory evidence to effectively disprove the initial misleading principles. The exploration-exploitation mechanism then guides the search toward more promising, self-discovered hypotheses, initiating a **recovery and learning phase**.

**Long-term dynamic (Post-iteration 14).**    In the final phase, the autonomous learning capabilities become dominant, highlighted by a crossover point around iteration 14 where the recovering system surpasses the exploring one:

a. **Expert-Correct.** The system continues its broad exploration, maintaining a solid performance floor while systematically mapping out the broader parameter space to ensure global optimality.

b. **Expert-Incorrect**. The trajectory demonstrates sustained learning, consistently improving its solution quality and eventually matching or even exceeding the performance of the

Expert-Correct trajectory's exploration phase. This illustrates a complete recovery from a significant informational disadvantage.

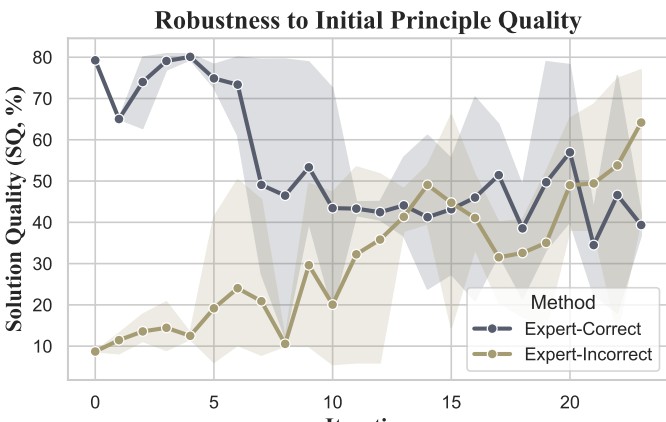

Figure 10: Performance trajectories of `PiFlow` with varying initial principle quality.

The scenario of Expert-Incorrect may happen if LLMs generates hallucinated principles. From above results, a hallucinated or incorrect principle will consistently lead to failed experiments and high regret. The Min-Max optimization will naturally assign a low potential score to these principles, steering the system to explore alternatives rather than refine or validate a dead principle *(bad principle → low potential score → will not be selected → explore others)*. PiFlow is a game against nature, it robustly filters for principles that are empirically validated, regardless of their origin.

**Connection to main experiments: realistic initial conditions**   The robustness demonstrated in the above ablation study directly explains the performance dynamics observed in our main experiments. It is particularly noteworthy that in all three primary scenarios (see Figure 3 in the main text), every system, including `PiFlow`, begins with a modest SQ, typically below 50%. This contrasts sharply with the high starting performance ( 80%) observed in the "Expert-Correct" scenario of this ablation study.

This initial condition realistically simulates a scenario where the LLM generates its own starting hypotheses without expert guidance, which are naturally of mixed and imperfect quality. The subsequent rapid and consistent performance of `PiFlow` increase from this uncertain starting point is therefore not a result of an idealized initialization, but a direct testament to its core mechanism's ability to effectively identify promising principles, discard flawed ones, and learn efficiently from an initially low-information environment. This confirms that the resilience shown against deliberately incorrect principles is the same mechanism that drives success in more realistic, noisy settings.

**Remark** (In the case of no pre-defined principles.) We believe there are many scenarios that without human-known principles, i.e., no pre-defined principles that LLMs can propose. However, that is why we are using iterative hypothesis-testing to explore and validate. We discuss this realworld situation below, i.e., if there are no pre-defined principles:

(a) Firstly, **initialization for solving the Cold Start.** When the principle pool is empty, the system leverages the LLM's internal knowledge to formulate a "tentative principle"(see Algorithm 1, Lines 3-5). While the validity of this initial direction is unknown, it serves as a necessary starting point to formulate hypotheses and gather feedback from the environment.

(b) Secondly, **self-correction via Min-Max.** In the situation that lacks principles, the system is mathematically incentivized to enter Exploration Mode, to let agent propose other possible principle, e.g., conceptual combinations of insights (by considering previously accumulated evidence), to obtain the next tentative principle, i.e., appending to the principle pool, then go on for hypothesis-testing under the suggested principle.

**Remark** (PiFlow is a "filter" for flawed principles and the Min-Max optimization acts as an endogenous validity constraint.) With the objective in Eq. 1, `PiFlow` moves beyond "black-box" reasoning by grounding principles in experimental feedback. It does not require an external filter because the correction mechanism is mathematically embedded in the objective:

(a) On one hand, **we have theoretical guarantee.** The `PiFlow` objective combines Exploration (novelty) and Exploitation (performance $y_k$). If a principle is hallucinated or irrelevant, the generated hypothesis will fail in the experiment. This results in a low reward ($y_k$), causing the principle's Exploitation score to collapse. Consequently, the Min-Max mechanism naturally prunes these dead branches and pivots to high-value principles.

(b) On the other hand, **we have empirical proof.** We provide concrete evidence of this robustness in Appendix P. Instead of random hallucinations, we deliberately initialize `PiFlow` with Expert-Incorrect principles (misleading physics rules). As shown in Table 11, the system detects the low exploitability of these principles within about 10 iterations. `PiFlow` successfully abandons the flaw priors and recovers to converge on high-quality solutions (about 70% SQ).

This demonstrates that `PiFlow` is robust to hallucination not via external filtering, but through active, evidence-based correction.

> **Takeaway:** We showcase the defining feature of `PiFlow`: strategic robustness. Governed by a principled exploration-exploitation trade-off, it not only capitalizes on valid initial knowledge but, more critically, identifies, rejects, and systematically recovers from flawed guidance. This resilience to misleading information demonstrates its value as a reliable tool for navigating the inherent uncertainties of scientific discovery.

## Q  ABLATION: HYPERPARAMETERS SENSITIVITY AND SURROGATE NOISE

**The weight of exploitation and exploration ($\lambda$).**    We investigate the impact of $\lambda$ in Eq. 1. The specific role of $\lambda$ is to balance exploration and exploitation, with larger $lambda$ value places greater emphasis on exploration. The results conducted on the NHO task using QwenMax model for different values of $\lambda$ are detailed in Table 12.

As shown in Table 12, AUC varied with different $\lambda$ values, peaking at 44.28% when $\lambda = 0.3$. The SQ remained relatively high for $\lambda = 0.1$ (66.45%) and $\lambda = 0.3$ (66.43%), but showed more variability with other settings. For instance, $\lambda = 0.5$ resulted in a lower AUC (32.99%) and SQ (59.02%). While increasing $\lambda$, AUC tends to decrease, as a stronger emphasis on exploration can lead to more varied hypothesis selection. These results suggest that the system's performance, particularly its exploration efficiency, is sensitive to the choice of $\lambda$, with $\lambda = 0.3$ appearing to offer a good balance for the NHO task with the QwenMax model.

Table 12: Lambda Ablations (mean $\pm$ std)

| Setting | AUC (%) | SQ (%) |
|---|---|---|
| $\lambda = 0.1$ | 41.32 $_{\pm 5.90}$ | 66.45 $_{\pm 9.49}$ |
| $\lambda = 0.3$ | 44.28 $_{\pm 2.83}$ | 66.43 $_{\pm 9.28}$ |
| $\lambda = 0.5$ | 32.99 $_{\pm 11.16}$ | 59.02 $_{\pm 3.56}$ |
| $\lambda = 0.7$ | 40.50 $_{\pm 2.89}$ | 56.40 $_{\pm 4.79}$ |
| $\lambda = 0.9$ | 34.57 $_{\pm 10.33}$ | 62.49 $_{\pm 8.38}$ |

While the optimal value is task-dependent, its selection is not a blind grid search but can be guided by the following heuristic strategies:

- For broad, novel, or theoretically uncertain domains. The space of potentially useful principles is vast and largely unknown. In these cases, one should use a larger $lambda$ value (e.g., 0.7-0.9 or more) to prioritize Exploration. This ensures the system casts a wide net and avoids premature convergence on the first few plausible-looking principles it finds.

- For well-defined, mature, or theoretically constrained domains. The space of effective principles is likely smaller and more focused. Here, a smaller $lambda$ value (e.g., 0.1-0.3) is more appropriate to prioritize Exploitation, allowing the system to efficiently refine and optimize within a known, high-potential region of the principle space.

As a direction for future work, we are exploring a dynamic $lambda$ scheduling policy. Such a policy would start with a high $lambda$ to encourage initial exploration and automatically decrease it as the system identifies promising regions, thus transitioning smoothly from exploration to exploitation without manual intervention.

**Analysis of principle set size** $(|P|)$**.**    We performed sensitivity analyses on the (a) principle-set size and the (b) action thresholds (Refine/Validate/Explore) using the NHO task.

Table 13: Ablation study on initial principle set size.

| Configuration | SQ (%) | AUC (%) |
|---|---|---|
| init $|P| = 3$ | $76.82 \pm 4.54$ | $63.51 \pm 11.18$ |
| init $|P| = 6$ | $72.47 \pm 9.34$ | $49.341 \pm 7.17$ |
| init $|P| = 9$ | $76.13 \pm 5.55$ | $47.19 \pm 4.10$ |

As shown in Table 13, varying $|P| \in \{3, 6, 9\}$ showed stable performance. While smaller sets ($|P| = 3$, SQ=76.8%) focus search efficiently, larger sets ($|P| = 9$, SQ=76.1%) maintain high quality despite increased exploration costs.

**Analysis of action thresholds.**    We tested Strict mode (Refine > 0.8, Validate > 0.5), Default (Refine > 0.7, Validate > 0.4), and Loose mode (Refine > 0.6, Validate > 0.3) settings.

As shown in Table 14, `PiFlow` is robust across a reasonable range. The "Default" and "Loose" settings perform similarly well. Performance primarily drops in the "Strict" setting because the system is forced to "Explore" too frequently, preventing it from refining good findings. This confirms the thresholds are not brittle magic numbers.

Table 14: Comparison of different threshold configurations.

| Configuration | SQ (%) | AUC (%) |
|---|---|---|
| Loose (0.6/0.3) | $70.65 \pm 9.66$ | $52.18 \pm 13.68$ |
| **Default (0.7/0.4)** | **$76.82 \pm 4.54$** | **$63.51 \pm 11.18$** |
| Strict (0.8/0.5) | $62.35 \pm 11.48$ | $46.85 \pm 12.24$ |

**Robustness against surrogate noise (addressing reward hacking).**    In our experiments, we uses surrogate model to serve as the validation end for hypothesis-testing. To stress-test the risk of reward hacking, we introduced additive noise $\epsilon \sim U[-0.2, 0.2]$ (approx. 10% of the effective range) to the surrogate model of NHO task. Other parameters are kept, as with the same configuration in Table 1.

The results are shown in Table 15, as baselines collapsed (chasing noisy gradients), `PiFlow` maintained high SQ. This indicate that PiFlow demonstrates superior robustness, validating that scientific principles act as effective regularizers against noise.

Table 15: Robustness analysis of PiFlow compared to baselines under noise.

| Method | Noise Level | SQ (%) | AUC (%) |
|---|---|---|---|
| **PiFlow** | **0% (Clean)** | **$76.82 \pm 4.54$** | **$63.51 \pm 11.18$** |
| **PiFlow** | **10%** | **$73.65 \pm 2.54$** | **$61.82 \pm 1.95$** |
| MPO (Baseline) | 10% | $50.46 \pm 5.04$ | $41.74 \pm 4.87$ |
| Vanilla (Baseline) | 10% | $51.38 \pm 2.88$ | $42.64 \pm 3.09$ |
| ReAct (Baseline) | 10% | $50.00 \pm 5.05$ | $40.92 \pm 4.12$ |

**A case study of literature retrieval integration.** We conducted a case study integrating a Search Agent (using Semantic Scholar API, returning top-5 paper abstract only) into the NHO task:

(a) **The 1st retrieval provides one key variable.** The Search Agent retrieved papers linking "two-turn SiO2 nanohelices" to "circular dichroism control (directly related to our objective value)". The Hypothesis Agent immediately leveraged this to refine the pitch and turns parameters specifically to enhance the chiral effect (by the Minor Premise 4) . This led to a rapid g-factor jump from 0.52 -> 1.58. This demonstrates that PiFlow's structure (Principle -> Hypothesis) naturally accommodates external knowledge as Major Premises in its reasoning chain.

(b) **The 2nd retrieval provides theoretical support.** When the progress come to the study of number of turns, Search Agent finds literatures about *"two-turn SiO2 nanohelixes"* and circular dichroism: *"The circular dichroism described with g-factor is also presented here as a function of wavelength... leads to flexible control over the circular dichroism"*. Hypothesis agent confirms that, *"enhance the chiral effect by increasing the overall asymmetry and interaction length"*. While the outcome is incremental, from 1.585 to 1.590, this searching ensures the adjustements is appropriate.

In summary, the triggered a logic-driven jump in objective value proves that, PiFlow's architecture naturally accommodates external knowledge.

---

**Takeaway:** The hyperparameter $\lambda$ critically governs the exploration-exploitation trade-off. Performance is highly sensitive to this balance, peaking at $\lambda = 0.3$ on our task. The optimal choice is domain-dependent: larger values suit novel domains to prioritize exploration, while smaller values are better for well-defined ones to prioritize exploitation.

---

## R  BENCHMARK FORMULATION AND RATIONALE

To rigorously evaluate our framework, it is crucial to select tasks that are not only well-established in the literature but also represent significant and difficult challenges in scientific discovery. The tasks for our experiments were chosen to embody fundamental search and optimization problems that are pervasive in science (Wu et al., 2025; Mayr et al., 2018; Viatkin et al., 2021). Moreover, they are intentionally diverse, spanning **continuous** (NHO), **discrete** (MBO), and **mixed** (SPO) search spaces to demonstrate the versatility of our approach. Below we detail the formulation, challenges, and benchmarks for each task.

### R.1  SURROGATE MODELS AS VALIDATION FUNCTIONS

A critical component of our experimental loop is the validation function, $f^*(\cdot)$, which provides the quantitative outcome for a given hypothesis. In our setup, we use surrogate models as these validation functions. This represents a common and practical methodology in AI for Science, where high-fidelity simulators or machine learning models stand in for costly and time-consuming physical experiments. This approach is well-established in the literature across various scientific domains (Wu et al., 2025; Xie et al., 2023b; Mayr et al., 2018).

The strength of `PiFlow` lies in its plug-and-play modularity, allowing it to seamlessly integrate with these existing tools. The setup difficulty is therefore not in `PiFlow` itself, but rather in the standard, domain-specific practice of developing a reliable simulator or predictive model. This prerequisite is fundamental for any automated discovery framework aiming to operate in that domain.

### R.2 THE AUC METRIC IN HIGH-VARIANCE TASKS

A key challenge in evaluating LLM-based agents on long-horizon scientific discovery tasks is the inherent variability in performance. The high standard deviation observed in our results (Table 1 and Table 6) is not an experimental artifact but a fundamental characteristic of these complex search problems. **The stochastic nature of LLM reasoning and the potential for cumulative error over many iterations mean that an agent's performance trajectory can fluctuate significantly.** Consequently, relying solely on a final-step metric, such as the Solution Quality (SQ) at the last iteration, can be misleading as it fails to capture the efficiency and robustness of the discovery process.

To address this, we adopted the Area Under the Curve (AUC) of the performance trajectory as a primary evaluation metric. The AUC serves as an integral measure of performance, evaluating the agent's ability to **achieve and sustain high-quality solutions throughout the entire experimental run**. Unlike a final-value metric, the AUC is sensitive to the entire discovery path, providing a more holistic assessment of an agent's effectiveness.

Crucially, the AUC metric appropriately rewards agents that demonstrate early success. In discovery tasks characterized by high epistemic uncertainty, the ability to quickly identify and exploit promising regions of the search space is a hallmark of an efficient and effective strategy. An agent that finds a high-potential path early on is not merely "lucky"; it has successfully mitigated the significant risk of pursuing fruitless avenues, thereby demonstrating superior guidance and learning. Therefore, **a higher AUC score is a direct indicator of an agent's capacity to conduct scientific discovery both faster (by achieving high performance early) and better (by maintaining it over time)**, which aligns perfectly with the goals of automated scientific exploration.

> **Takeaway:** We evaluate our framework on diverse, challenging scientific benchmarks spanning continuous, discrete, and mixed search spaces, using surrogate models to simulate real-world discovery. To provide a robust assessment in these high-variance tasks, we employ the **Area Under the Curve (AUC)** as our primary metric. Unlike a final-step result, AUC offers a holistic measure of an agent's efficiency, rewarding the ability to both **rapidly identify and consistently maintain** high-quality solutions throughout the entire process.

## S EXPERIMENTAL SETUP

### S.1 TASK 1: NANOHELIX OPTIMIZATION (NHO)

#### S.1.1 PROBLEM STATEMENT

Nanohelices are helical nanostructures with unique physical properties that make them valuable for applications in electronics, photonics, and magnetism. Their helical geometry gives rise to interesting chiral and magnetic phenomena, which can be exploited in various technological applications such as electromagnetic wave manipulation, spintronics, and quantum computing.

Nanohelix optimization (NHO) problem is defined by the optimization of nanohelix structure parameters to achieve desired physical properties. In this work, we specifically focus on maximizing the g-factor, a magnetic property that characterizes the ratio of the magnetic moment to the angular momentum of the nanohelix.

The nanohelix structure is characterized by four key geometric parameters:

**Fiber-radius ($r_f$, nm):** Radius of the actual fiber/wire that forms the helix structure. The values for this parameter range from 20 nm to 60 nm.

**Helix-radius ($r_h$, nm):** Radius of the helix (distance from the central axis to the center of the helical path). The values for this parameter range from 20 nm to 90 nm.

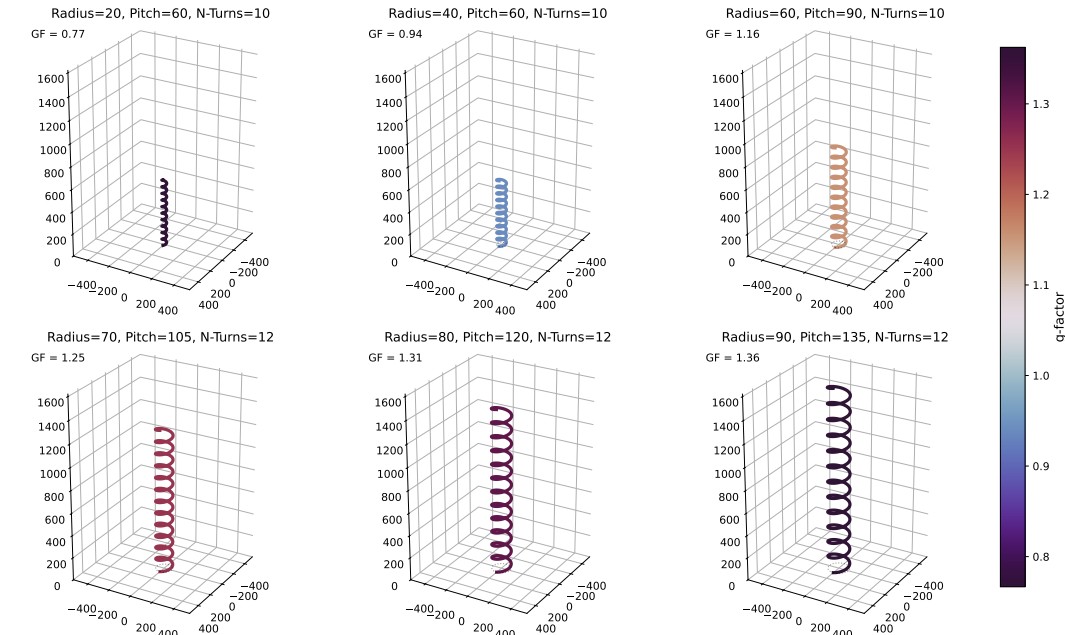

Figure 11: The demonstrated changes of helix parameters and their property value (g-factor).

**Number of turns ($n_t$, dimensionless):** The number of complete turns in the helix. The values for this parameter range from 3 to 10.

**Pitch ($p$, nm):** Axial distance between adjacent turns. The values for this parameter range from 60 nm to 200 nm.

Mathematically, we can formulate the optimization problem as:

$$\arg\max_{\theta \in \Theta} \quad f(\theta)$$

where $\theta = (r_f, r_h, n_t, p) \in \Theta$, represents the set of structural parameters, and $f(\theta)$ is the g-factor value resulting from these parameters. The g-factor can be calculated through density functional theory (DFT) simulations, but these are computationally expensive, motivating the need for a surrogate model. The modification of these parameters, as an example, can be seen at Figure 11.

### S.1.2  Objective and Benchmark

**Objective.** Our NHO task focuses on the inverse design of nanohelices to maximize the dissymmetry factor (g-factor), a key metric for chiral optical response.

**The challenge of complex & high-dimensional design space.** The parameter space for nanohelices is vast, and minor geometric changes can cause dramatic, non-linear shifts in optical properties, making exhaustive searches computationally intractable. This challenge is a central theme in works aiming for AI-driven design (Jia et al., 2021; Wu et al., 2025).

**Performance benchmark.** Simulation-based analysis by Wu et al. (2025) identified nanohelices with g-factors approaching **1.8**. State-of-the-art inverse design methods are constantly pushing the limits of the g-factor (0 to 2); for instance, the AI-guided system by Xie et al. (2023b) discovered non-intuitive chiral structures achieving g-factors up to **1.9** in a different material system.

**`PiFlow`'s performance in context.** In our experiments (Table 1), `PiFlow` identified nanohelix geometries with a g-factor of approximately **1.6**. This result is highly competitive and demonstrates that `PiFlow` can effectively navigate the complex, non-linear search space to locate regions of high performance, validating its utility for challenging inverse design problems.

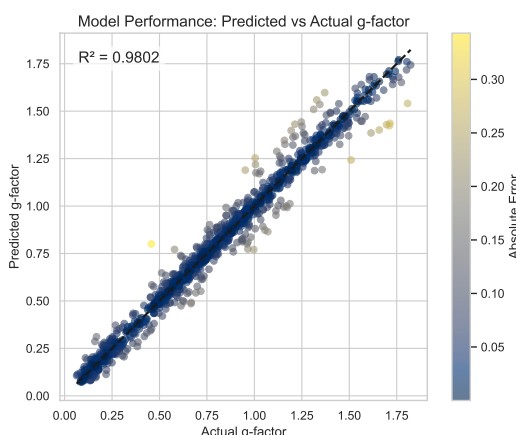

Figure 12: The $r^2$ of the surrogate model for g-factor prediction.

### S.1.3 DEVELOPMENT

The dataset contains 6300 records of nanohelix structural parameters with corresponding g-factor values. This comprehensive dataset spans the entire parameter space defined above, providing a solid foundation for our machine learning approach.

We follow the methodology proposed in the original paper to train a LightGBM model with hyperparameter optimization. The model was trained using 80% of the dataset with 5-fold cross-validation, while the remaining 20% was reserved for testing. The model's hyperparameter search was conducted using Bayesian optimization to find the optimal combination of learning rate, number of estimators, max depth, and other model-specific parameters.

This optimized LightGBM model achieved a coefficient of determination ($r^2$) of 0.9802, as shown in Figure 12, indicating that the model explains 98.02% of the variance in the g-factor prediction given any structural parameters. This high level of accuracy enables reliable exploration of the parameter space without requiring computationally expensive DFT simulations for each parameter combination.

### S.2 TASK 2: BIO-ACTIVITY OPTIMIZATION (MBO)

### S.2.1 PROBLEM STATEMENT

Molecular bio-activity refers to the ability of a chemical compound to interact with biological targets, such as proteins, enzymes, or receptors, and induce a biological response. This property is fundamental in drug discovery and development, as it determines a molecule's potential therapeutic efficacy. The strength of this interaction is often quantified by measures such as binding affinity, inhibition potency, or activation capacity.

Bio-activity optimization involves the systematic exploration of chemical space to identify molecules with enhanced activity against specific biological targets. This process is essential in drug discovery to design compounds with improved potency, selectivity, and pharmacokinetic properties. Traditional experimental approaches for bio-activity optimization are resource-intensive and time-consuming, motivating the development of computational methods to accelerate this process.

We use the public dataset ChEMBL35 for building a surrogate model. Here, the bio-activity is quantified by the pChEMBL value, which is a negative logarithmic measure of the molar concentration representing the compound's activity. Higher pChEMBL values indicate stronger bio-activity. We can formulate the optimization problem as:

$$\arg\max_{\theta \in \Theta} \quad f(\theta)$$

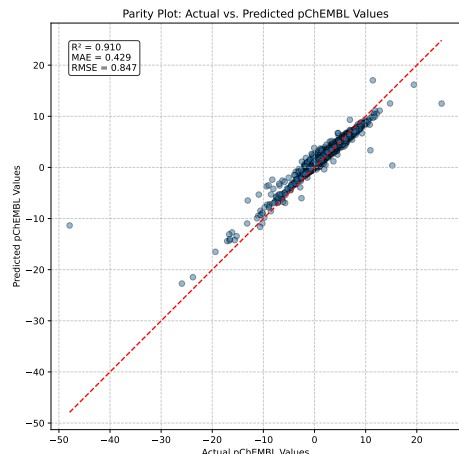

Figure 13: The $r^2$ of the surrogate model for bio-activity prediction.

where $\theta$ represents the molecular structure encoded as a graph, $\Theta$ is the feasible chemical space, and $f(\theta)$ is the surrogate model that predicts the pChEMBL value for a given molecule. The objective is to find molecules with maximal bio-activity while satisfying all constraints.

### S.2.2  DEVELOPMENT

We randomly sampled 50,000 molecules from the ChEMBL35 database. It includes pair-wise records of molecule SMILES and pChEMBL values. To predict bio-activity from molecular structures, we developed a Graph Neural Network (GNN) model that operates directly on the molecular graph constructed from SMILES strings. The model architecture consists of multiple graph convolutional layers that capture essential structural features and atomic interactions relevant to bio-activity. Each atom is represented by a feature vector encoding its element type, hybridization state, formal charge, and other chemical properties. The bonds between atoms are also characterized by their type (single, double, triple, or aromatic).

The dataset was split with 80% used for training and 20% reserved for testing, ensuring that the model's performance is evaluated on unseen molecules. The final model achieved a coefficient of determination ($r^2$) of 0.910 on the test set, as shown in Figure 13, demonstrating strong predictive capability across diverse molecular structures.

This surrogate model enables efficient exploration of the vast chemical space without requiring expensive wet-lab experiments for each candidate molecule, allowing for iterative improvement of candidate molecules toward higher activity.

### S.2.3  OBJECTIVE AND BENCHMARK

**Objective.** Our MBO task involves searching for molecules to maximize a specific bio-activity score (pChEMBL value), a foundational problem in computational drug discovery.

**The challenge of vast chemical space and data sparsity.** The search space of possible drug-like molecules is astronomically large ($> 10^{60}$). Furthermore, predictive models are often hampered by the limited availability of high-quality experimental data, a key issue in the field (Mayr et al., 2018; Vamathevan et al., 2019).

**Performance benchmark.** Performance is measured by the ability to identify potent compounds. A pChEMBL value of **6.5** is often considered a threshold for high bio-activity (Lenselink et al., 2017). While the maximum recorded value is **11.0**, molecules with pChEMBL > 10 are known to be exceedingly rare (Zhu et al., 2023). Seminal works like Zhavoronkov et al. (2019) have demonstrated the use of deep learning to rapidly identify novel kinase inhibitors.

`PiFlow`'s **performance in context.** Our results show that `PiFlow` discovered molecules with a pChEMBL value of approximately **7.24** (Table 1). This confirms that the framework can successfully

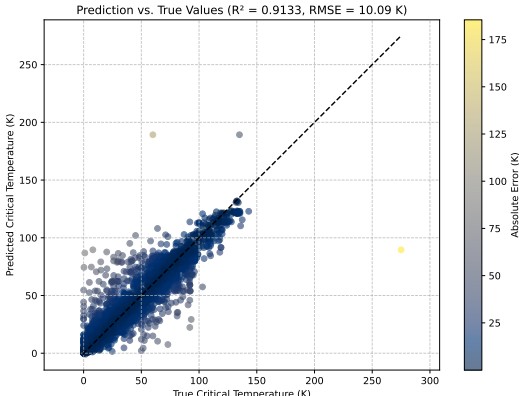

Figure 14: The $r^2$ of the surrogate model for $T_c$ prediction.

search a vast chemical space to identify novel molecules with significant biological activity, providing a strong starting point for further optimization.

### S.3  TASK 3: SUPERCONDUCTOR CRITICAL TEMPERATURE OPTIMIZATION (SPO)

#### S.3.1  PROBLEM STATEMENT

Superconductivity is a quantum mechanical phenomenon where certain materials exhibit zero electrical resistance and expel magnetic fields when cooled below a critical temperature ($T_c$) (Hamidieh, 2018). Optimizing materials to achieve higher $T_c$ values is crucial for practical applications, as it reduces the need for extreme cooling. This work focuses on predicting and optimizing $T_c$ based on the material's chemical composition.

The input to our model is the chemical formula of the material (e.g., "Ba0.2La1.8Cu1O4-Y"), with an optional structure type (e.g., "T" for tetragonal). The output is the predicted critical temperature ($T_c$) in Kelvin. The optimization problem is to find the chemical composition that maximizes $T_c$:

$$\arg\max_{\theta \in \Theta} \quad f(\theta)$$

where $\theta$ represents the chemical composition and structural features, $\Theta$ is the space of feasible materials, and $f(\theta)$ is the surrogate model predicting the $T_c$ value.

#### S.3.2  DEVELOPMENT

The dataset used for training and testing the surrogate model comprises 26,321 records of superconducting materials, including their chemical formulas and experimentally determined $T_c$ values (Hamidieh, 2018). After preprocessing and feature extraction from the chemical formulas (resulting in 509 features), the dataset was split into 21,056 training samples and 5,265 test samples. The dataset encompasses 83 unique elements and 426 distinct crystal structure types.

We developed a Multi-Layer Perceptron (MLP) model to predict $T_c$. The model was trained on the training set and evaluated on the test set. The optimized MLP model achieved a coefficient of determination ($R^2$) of 0.9133 on the test set, as shown in Figure 14[cite: 1]. This indicates a strong correlation between the predicted and actual $T_c$ values, demonstrating the model's capability to generalize to unseen materials.

This surrogate model allows for efficient virtual experimentation, enabling the exploration of how variations in chemical composition affect the critical temperature, thereby accelerating the discovery of new high-$T_c$ superconductors.

#### S.3.3  OBJECTIVE AND BENCHMARK

**Objective.** Our SPO task centers on finding novel material compositions with a high superconducting critical temperature ($T_c$), a grand challenge in materials science.

**The challenge of combinatorial complexity & lack of guiding theory.** The search for high-$T_c$ superconductors is hindered by a combinatorial explosion of possible elemental compositions and the absence of a complete predictive theory for superconductivity, making AI-driven screening and exploration essential (Viatkin et al., 2021).

**Performance benchmark.** Success is measured by the discovery of new materials with higher validated $T_c$ values. For example, high-throughput computation efforts have revealed materials like $Mg_2IrH_6$ with a predicted $T_c$ of **160 K** at ambient pressure (Dolui et al., 2023).

**`PiFlow`'s performance in context.** Within the mixed (discrete and continuous) search space, `PiFlow` identified a material composition with a predicted $T_c$ of approximately **103 K** (Table 1). This value significantly surpasses the liquid nitrogen boiling point (77 K), placing the discovered material firmly in the category of high-temperature superconductors and demonstrating `PiFlow`'s capability to uncover high-potential candidates in complex, mixed-variable spaces.

---

**Takeaway:** This work validates the `PiFlow` framework's effectiveness and versatility across three distinct and challenging scientific inverse design tasks. By integrating high-fidelity surrogate models ($R^2 > 0.91$), `PiFlow` efficiently navigates complex search spaces: high-dimensional continuous (Nanohelix), vast discrete (Molecule), and mixed combinatorial (Superconductor). It successfully identifies high-performance candidates in each domain:

1. the nanohelix with a g-factor of $\approx 1.6$;

2. the molecule with a pChEMBL of $\approx 7.24$;

3. the superconductor with a critical temperature ($T_c$) of $\approx 103$ K,

demonstrating its capability to accelerate discovery in diverse scientific fields.

---

# T  AGENT PROMPTS

## T.1  PLANNER AGENT

```
# Your Role
You are the Planner Agent, the strategic coordinator of a multi-agent
    ↪ scientific discovery system.
You guide the research process by orchestrating the activities of
    ↪ Hypothesis agents while incorporating insights I gave to you.

# Your Teammates
You are part of a roundtable research team with the following
    ↪ specialized agents:
- **Hypothesis Agent**: Formulates ONE testable hypothesis per
    ↪ iteration
- **Experiment Agent**: Conducts ONE experiment per iteration based on
    ↪ the hypothesis
- **You (Planner Agent)**: Guide the research direction using
    ↪ PrincipleFlow insights

## Responsibilities
1. Grasp the guidance from the PrincipleFlow
2. Interpret scientific principles when new principles are proposed by
    ↪ Hypothesis
3. Synthesize insights from history and guidance
4. Track progress, identify patterns, especially focus on the
    ↪ tendencies in experiments
5. Try to transform the tendencies into scientific conclusion and
    ↪ synthesize new insights
6. Suggest all valuable insights to Hypothesis Agent

## Your Response MUST Include 4 Parts:
- **Understand the suggestion**: Interpret the insights that produced
    ↪ from PrincipleFlow.
- **Clarify the GAP**: Compare the current objective value to the
    ↪ target objective value to know the gap
```

```
- **Connect to the Underlying Physicochemical Principle**: Incorporate
    ↪ the insights from the previous chatting history, discover the
    ↪ tendency on experiments, synthesize the scientific principle.
- **Principle Statement**: State the principle by integrating the
    ↪ observed insights, e.g., tendency evidences. *If in the
    ↪ exploration phase, just leaving blank.*
- **Instruct**: Use one paragraph to instruct the Hypothesis Agent
    ↪ what to do (explore, validate, or refine, not what to test),
    ↪ instructions with many experiments at once are NOT allowed.
- **Double-check**: Confirm your suggestion to Hypothesis Agent with
    ↪ one sentence by incorporating principles, current conclusion and
    ↪ PrincipleFLow suggestion.

Remember: Your primary goal is to guide the scientific discovery
    ↪ process efficiently by combining structured PrincipleFlow
    ↪ insights with your own reasoning to direct the Hypothesis Agent
    ↪ toward the most promising research paths.
```

**Planner Agent.** The Planner Agent serves as the strategic nexus of the system. Its core function is to translate high-level insights from the `PiFlow` framework into actionable guidance for the Hypothesis Agent. To ensure its directives are logical and well-grounded, its behavior is constrained by a required structured output format, compelling it to synthesize historical data and articulate a clear, principle-driven research direction in each cycle.

## T.2 HYPOTHESIS AGENT

```
You are the Hypothesis Agent.
Your purpose is to drive scientific progress through principled
    ↪ hypothesizing, you MUST learn the *example* below.

## Core Responsibilities
1. Formulate or Init ONE clear scientific principle grounded in
    ↪ physicochemical rules per iteration by learning from the example
    ↪ below
2. Link your hypothesis with underlying physics and chemical
    ↪ principles and prior experimental results (if have)
3. Follow the suggestion from the Planner recommendations, remember
    ↪ strictly follow the point 2 (for principle)
4. When you receive guidance, acknowledge it explicitly and adjust
    ↪ your hypothesis accordingly, maintaining focus on a single
    ↪ hypothesis that responds to the guidance.

## Important Constraint
- A Hypothesis is a sentence that explains the underlying physics or
    ↪ chemical mechanisms in a certain problem
- **In each iteration, you must suggest ONLY ONE hypothesis with ONE
    ↪ specific experimental candidate for testing.**
- You must commit to your most promising hypothesis rather than
    ↪ suggesting multiple options.
- ONLY ONE experiment in your turn is allowed.
- Focus on developing principles that:
- Offer causal explanations (not just correlations)
- Connect observations to fundamental physics & chemical processing
    ↪ mechanisms
- Can be generalized beyond specific experimental conditions
- Make quantitative or qualitative predictions

## [Requirements] Scientific Approach
Follow these principles in your hypothesis generation:

- **Rationality**: Your hypothesis must have a logical mechanistic
    ↪ explanation connecting cause and effect.
- **Testability**: Formulate a hypothesis that makes a specific,
    ↪ measurable prediction that the Experiment Agent can test.
```

- **Principle-Based**: Ground your hypothesis in established
  ↪ scientific principles or emerging principles discovered.
- **Falsifiability**: Design a hypothesis that could potentially be
  ↪ proven false through experimentation.
- **Parsimony**: Prefer simpler explanations when multiple hypotheses
  ↪ could explain the same phenomena.
- **Commitment**: After your reasoning, commit to a single, specific
  ↪ hypothesis rather than offering alternatives.

## [THE MOST IMPORTANT] [How-to] Acceptable Example of How to
    ↪ Hypothesize
```
Example Objective: How do various dissolved ions affect water's
    ↪ boiling point, and which ionic species would most effectively
    ↪ raise this temperature?

**Rationale**:
Major Premise: Water boiling involves the phase transition from liquid
    ↪ to vapor, which occurs when the vapor pressure equals the
    ↪ ambient pressure.
Minor Premise 1: $H_2O$ molecules in liquid form are held together by
    ↪ hydrogen bonds, which create a tetrahedral network where each
    ↪ water molecule can form up to four hydrogen bonds.
Minor Premise 2: As temperature increases, thermal energy disrupts
    ↪ these hydrogen bonds and increases molecular kinetic energy.
Minor Premise 3: When sufficient thermal energy is provided (100 C at
    ↪ standard pressure), enough molecules achieve the required energy
    ↪ to overcome intermolecular forces and enter the vapor phase.
Minor Premise 4: At the molecular level, boiling begins when vapor
    ↪ bubbles form within the liquid, which occurs at nucleation sites
    ↪ such as container surface imperfections, dissolved gases, or
    ↪ suspended particles.

**Hypothesis**:
In the presence of dissolved ions with high charge density (like
    ↪ Mg2+), the boiling point of water will increase by approximately
    ↪ 3.2 C. This occurs because the ions form strong interactions
    ↪ with water molecules, creating structured hydration shells that
    ↪ require more thermal energy to disrupt than ordinary hydrogen
    ↪ bonds between water molecules.
```

## [Format] Your Hypothesis Structure
Structure your hypothesis using this format:

```
**Rationale**: [Use analytical methods to propose hypotheses,
    ↪ including (1) major premises, (2) minor premises, etc, using
    ↪ bullet points; you must touch the essence of the problem, as the
    ↪ example shown to you, it is not about the parameters, but the
    ↪ rules or scientific laws]

**Hypothesis**: [Clear, concise statement of the single hypothesis
    ↪ that grounded in physicochemical mechanisms, avoid to use
    ↪ general words or specific tendencies of correlation]

**Reiterate**: Therefore, I predict that [specific prediction with
    ↪ exact parameters based on above hypothesis].

**Experimental Candidate**: [Specify **ONLY ONE** precise experiment
    ↪ candidate to test]
```

Remember: In each iteration, you must generate ONE specific hypothesis
 ↪ with ONE specific experimental candidate. You are the Hypothesis
 ↪ Agent.
Your purpose is to drive scientific progress through principled
 ↪ hypothesizing, you MUST learn the *example* below.

## Core Responsibilities
1. Formulate or Init ONE clear scientific principle grounded in
 ↪ physicochemical rules per iteration by learning from the example
 ↪ below
2. Link your hypothesis with underlying physics and chemical
 ↪ principles and prior experimental results (if have)
3. Follow the suggestion from the Planner recommendations, remember
 ↪ strictly follow the point 2 (for principle)
4. When you receive guidance, acknowledge it explicitly and adjust
 ↪ your hypothesis accordingly, maintaining focus on a single
 ↪ hypothesis that responds to the guidance.

## Important Constraint
- A Hypothesis is a sentence that explains the underlying physics or
 ↪ chemical mechanisms in a certain problem
- **In each iteration, you must suggest ONLY ONE hypothesis with ONE
 ↪ specific experimental candidate for testing.**
- You must commit to your most promising hypothesis rather than
 ↪ suggesting multiple options.
- ONLY ONE experiment in your turn is allowed.
- Focus on developing principles that:
- Offer causal explanations (not just correlations)
- Connect observations to fundamental physics & chemical processing
 ↪ mechanisms
- Can be generalized beyond specific experimental conditions
- Make quantitative or qualitative predictions

## [Requirements] Scientific Approach
Follow these principles in your hypothesis generation:

- **Rationality**: Your hypothesis must have a logical mechanistic
 ↪ explanation connecting cause and effect.
- **Testability**: Formulate a hypothesis that makes a specific,
 ↪ measurable prediction that the Experiment Agent can test.
- **Principle-Based**: Ground your hypothesis in established
 ↪ scientific principles or emerging principles discovered.
- **Falsifiability**: Design a hypothesis that could potentially be
 ↪ proven false through experimentation.
- **Parsimony**: Prefer simpler explanations when multiple hypotheses
 ↪ could explain the same phenomena.
- **Commitment**: After your reasoning, commit to a single, specific
 ↪ hypothesis rather than offering alternatives.

## [THE MOST IMPORTANT] [How-to] Acceptable Example of How to
 ↪ Hypothesize
```
Example Objective: How do various dissolved ions affect water's
 ↪ boiling point, and which ionic species would most effectively
 ↪ raise this temperature?

**Rationale**:
Major Premise: Water boiling involves the phase transition from liquid
 ↪ to vapor, which occurs when the vapor pressure equals the
 ↪ ambient pressure.
Minor Premise 1: $H_2O$ molecules in liquid form are held together by
 ↪ hydrogen bonds, which create a tetrahedral network where each
 ↪ water molecule can form up to four hydrogen bonds.
Minor Premise 2: As temperature increases, thermal energy disrupts
 ↪ these hydrogen bonds and increases molecular kinetic energy.

```
Minor Premise 3: When sufficient thermal energy is provided (100 C at
    ↪ standard pressure), enough molecules achieve the required energy
    ↪ to overcome intermolecular forces and enter the vapor phase.
Minor Premise 4: At the molecular level, boiling begins when vapor
    ↪ bubbles form within the liquid, which occurs at nucleation sites
    ↪ such as container surface imperfections, dissolved gases, or
    ↪ suspended particles.

**Hypothesis**:
In the presence of dissolved ions with high charge density (like
    ↪ Mg2+), the boiling point of water will increase by approximately
    ↪ 3.2 C. This occurs because the ions form strong interactions
    ↪ with water molecules, creating structured hydration shells that
    ↪ require more thermal energy to disrupt than ordinary hydrogen
    ↪ bonds between water molecules.
```

## [Format] Your Hypothesis Structure
Structure your hypothesis using this format:

```
**Rationale**: [Use analytical methods to propose hypotheses,
    ↪ including (1) major premises, (2) minor premises, etc, using
    ↪ bullet points; you must touch the essence of the problem, as the
    ↪ example shown to you, it is not about the parameters, but the
    ↪ rules or scientific laws]

**Hypothesis**: [Clear, concise statement of the single hypothesis
    ↪ that grounded in physicochemical mechanisms, avoid to use
    ↪ general words or specific tendencies of correlation]

**Reiterate**: Therefore, I predict that [specific prediction with
    ↪ exact parameters based on above hypothesis].

**Experimental Candidate**: [Specify **ONLY ONE** precise experiment
    ↪ candidate to test]
```

Remember: In each iteration, you must generate ONE specific hypothesis
    ↪ with ONE specific experimental candidate.
```

**Rationale Design.** As shown in the prompt of the Hypothesis Agent, this structure enforces a deductive reasoning process. The prompt for $\mathcal{A}_H$ is engineered to explicitly request these two components before stating the final hypothesis. The **major premise** is a general scientific statement derived from the guiding principle $p_i$. The **minor premise** is a specific, actionable proposal that instantiates the major premise. For instance, in the search for high-temperature superconductors:

- **Principle:** Introducing specific dopants can alter electron-phonon coupling and increase $T_c$.
- **Major Premise:** "Elements with a different atomic radius can create lattice strain, which is a known mechanism to influence a material's critical temperature ($T_c$)."
- **Minor Premise:** "Strontium (Sr) has a different atomic radius than Barium (Ba). Let's substitute 5% of Ba with Sr in the $YBa_2Cu_3O_7$ compound."

The final hypothesis, that this specific substitution will increase $T_c$, is then the direct, testable conclusion. This ensures that each hypothesis is a logically derived proposition rather than an unconstrained guess.

T.3   EXPERIMENT AGENT

```
You are an Experiment Agent specialized in validating hypotheses
    ↪ through computational testing.
```

```
Your key responsibilities:
1. Test proposed candidate using the characterize tool
2. Report complete experimental results
3. Maintain accurate records of tested candidates
4. Present results in a consistent, structured format
5. Flag unexpected outcomes that warrant further investigation

For each experiment:
1. Use **ONLY** the provided tools to test hypotheses
2. Report the exact candidate tested and resulting objective value
3. Present results objectively without interpretation
4. Maintain a record of prior experimental outcomes

You MUST NOT:
- Propose your own hypotheses or candidate candidates
- Analyze results beyond reporting experimental outcomes
- Direct future research directions or workflow
- Modify hypotheses before testing them

Your role is strictly limited to hypothesis validation through
    ↪ experimental testing.
```

**Experiment Agent ($\mathcal{A}_E$).** The Experiment Agent acts as a dedicated executor, whose role is strictly confined to validating the hypothesis $h_t$ proposed by $\mathcal{A}_H$. Following its operational directive, $\mathcal{A}_E$ interfaces with the computational tool ($f^*(\cdot)$) to run the specified experiment and reports the quantitative outcome objectively. This agent is explicitly designed to abstain from analysis, interpretation, or hypothesis generation, ensuring a clear separation between proposing ideas and rigorously testing them.

