# OpenReview forum: "PiFlow: Principle-aware Scientific Discovery with Multi-Agent Collaboration"
_ICLR.cc/2026/Conference — Submitted to ICLR 2026_

### Official Review · Reviewer_TzZQ · 2025-10-16

**Soundness:** 3
**Presentation:** 2
**Contribution:** 3
**Rating:** 6
**Confidence:** 2

**Summary:**

The paper introduces PiFlow, a principle-driven framework based on multi-agent systems (MAS) designed to address key challenges in AI-driven scientific discovery—such as blind hypothesis generation, weak chains of evidence, and poor cross-domain generalization. Its core approach uses a min–max optimization to balance exploration (maximizing information gain) and exploitation (minimizing regret), and it leverages scientific principles (e.g., physical and chemical laws) to structurally reduce uncertainty.

**Strengths:**

1. Proposes PiFlow, a discovery framework based on multi-agent collaboration and information theory. It balances exploration (maximizing information gain) and exploitation (minimizing regret) via Min–Max optimization, and provides theoretical guarantees with an $O(\sqrt{T})$ regret bound.

2. Uses plug-and-play modules (e.g., Planner, Hypothesis, Experiment Agent). Validated across three domains—nanomaterials (g-factor ≈ 1.6), biomolecules (pChEMBL ≈ 7.24), and superconductors ($T_c$ ≈ 103 K)—with high surrogate model accuracy ($R^2>0.91$).

3. Demonstrates a 73.55% improvement in exploration efficiency (AUC) and a 94.06% improvement in solution quality (SQ); computational cost is only 1.5%, and token consumption is reduced by up to 27%.

**Weaknesses:**

1. Decision mechanism complexity of $O(t^2 \cdot d)$ may become a bottleneck. The framework depends on LLM-generated principles (e.g., QwenMax) and surrogate models; data biases (e.g., insufficient active molecules in ChEMBL35) could affect outcomes.

2. Some evaluations are simulated rather than experimentally measured (e.g., nanospiral g-factor). The paper lacks comparisons to baselines such as genetic algorithms or reinforcement learning and overlooks practical constraints (e.g., drug ADMET properties, superconducting crystal defects).

3. Early performance depends on the quality of initial principles (incorrect principles can yield SQ < 20%). Scalability in high-dimensional or dynamic spaces is not validated.

**Questions:**

See weakness.

---

> ### Author Response · Authors · 2025-11-21
>
> We thank the reviewer for recognizing PiFlow's **novelty**, **theoretical guarantees**, and **efficiency**, as well as its potential across multiple domains. Given the reviewer's confidence score (2) and the "marginally above acceptance" rating, we believe clarifying the technical misunderstandings below will strongly solidify the assessment of the paper's soundness and contribution.
>
> &nbsp;
>
> ### **[W1] Clarification on complexity and data bias.**
>
> ---
>
> **The complexity is negligible in practice, and PiFlow is designed specifically to mitigate data bias via physical principles.**
>
> - **Computational complexity (not a bottleneck).** The decision complexity $O(t^2 \cdot d)$ does not create a bottleneck because $t$ (iterations) is typically small in scientific discovery ($t < 50$).
>     - **We profiled the runtime in the NHO task.** Over 24 iterations, the entire PiFlow framework accounted for only **9.4%** of the total PiFlow-MAS runtime, while the surrogate model took **0.2%**.
>     - **We also quantitatively profiled the optimization efficiency** in the NHO task. The result confirms that the algorithmic complexity of PiFlow translates into superior search efficiency:
>         - For baselines, Vanilla Agent System achieves **SQ=50%** taking **2416s;** The other baseline MPO achieves **SQ=52%** taking **506s** and baseline ReAct even fails to reach such level of SQ in all its 918s.
>         - However, our PiFlow achieves **SQ=52%** by only **427s,** delivering such a result using less than **1/5 (~17.6%)** of the Vanilla Agent's time **(a 5.6x speedup)** and is **1.2x faster** than the baseline MPO.
>     - **Conclusion.** The vast majority of time is consumed by LLM inference. The specific similarity calculation ($t^2$) takes milliseconds, which is structurally distinct from—and negligible compared to—LLM generation (seconds) or real-world wet-lab experiments (hours/days).
> - **Robustness to data bias (strategic optimization).** While biases exist in datasets like ChEMBL, PiFlow‘s core contribution is a **principle-aware framework** that operates above the raw data layer.
>     - **Regularization via principles.** While biases exist in datasets like ChEMBL, PiFlow operates as a principle-aware framework above the raw data layer. We define regularization here as the mechanism that **penalizes semantic invalidity and constraints the search space** to chemically plausible regions, preventing the model from overfitting to biased local optima.
>     - **Results.** Our experiment explicitly demonstrates this mechanism:
>         - When the agent stagnated at pChEMBL=**5.58**, it began overfitting to the local bias by attempting minor, ineffective variations on a Thiophene scaffold. Our **PiFlow explicitly guiding the search** toward Indole rings to exploit electron-donating properties. This strategic correction immediately lifted the score to **6.44 (~13% increase of SQ)**. Furthermore, when the Indole scaffold saturated ($\sim$6.46)**, PiFlow corrected the search path**, ultimately achieving a global optimum of **7.65 (further ~18% SQ increase after 6.44)**, far exceeding the target of 6.5 (100% of SQ).
>     - **Conclusion.** The results confirm PiFlow's efficiency, while its principle-driven design effectively acts as a regularizer, ensuring robust discovery even when the underlying data is imperfect or biased.
>
> &nbsp;
>
> ### **[W2] Numerical baselines, real-world constraints, and simulated evaluation.**
>
> ---
>
> **PiFlow addresses the challenge of semantic hypothesis generation, where GA/RL face significant barriers in representation.**
>
> - **Clarification of why not GA/RL.** GA and RL typically require predefined, fixed state/action spaces and extensive hyperparameter tuning for specific tasks. In contrast, PiFlow operates in an open-ended **semantic space** (natural language principles), allowing it to be Plug-and-Play without domain-specific encoding.
> - **Practical constraints.** Practical constraints can be easily integrated into PiFlow's reward function $f^*(h_t)$ within the Min-Max objective. We focused on single-objective tasks to cleanly validate the theoretical framework, but the architecture naturally supports constrained optimization.
> - **Simulated rather than experimentally measured (e.g., nanohelix g-factor).** Using high-fidelity surrogates is a standard evaluation protocol in ML for Science to ensure reproducibility and benchmark consistency [1, 2, 3].
>
> **Conclusion:** PiFlow uniquely solves the **semantic search** problem where numerical baselines (GA/RL) fail, and its modular design provides the flexibility to handle real-world constraints (e.g., ADMET) without requiring architectural changes.

---

> ### Author Response · Authors · 2025-11-21
>
> ### **[W3] Dependence on initial principles.**
>
> ---
>
> **This result is actually strong evidence of PiFlow's self-correction capability, not a weakness.**
>
> We believe there is a **misunderstanding** regarding the results in Appendix P (Figure 10).
>
> - **Clarification of Fig. 10/11.** The reviewer notes that incorrect principles yield SQ < 20%. This refers to the starting point of our ablation study, where we intentionally injected false principles to test robustness.
> - **Proof of Robustness.** As demonstrated in **Figure 11**, PiFlow does **not** stay at 20%.
>     - **Mechanism of recovery from only 20% SQ initially:** The Min-Max mechanism successfully detects the conflict between the false hypothesis and the experimental feedback (Regret). Therefore, the Planner agent explicitly **discards the bad principles** and autonomously learns new, correct ones, eventually matching the performance of the Expert-Correct trajectory.
>
> **Conclusion:** This confirms that PiFlow is **resilient to human error**, a critical feature for AI-driven discovery where initial human intuition may be flawed.
>
> &nbsp;
>
> We hope these clarifications demonstrate PiFlow's robustness and efficiency, and we are happy to answer any further questions.
>
> &nbsp;
>
> ## Reference
>
> [1] Bursch, Markus et al. “Best‐Practice DFT Protocols for Basic Molecular Computational Chemistry.” *Angewandte Chemie (International Ed. in English)* 61 (2022): n. pag.
>
> [2] Koscher, Brent A. et al. “Autonomous, multiproperty-driven molecular discovery: From predictions to measurements and back.” *Science* 382 (2023): n. pag.
>
> [3] Stanev, Valentin G. et al. “Machine learning modeling of superconducting critical temperature.” *npj Computational Materials* 4 (2017): 1-14.

---

### Official Review · Reviewer_bk9d · 2025-10-29

**Soundness:** 2
**Presentation:** 2
**Contribution:** 2
**Rating:** 4
**Confidence:** 4

**Summary:**

This paper introduces PiFlow, an information-theoretic framework designed to enhance automated scientific discovery using large language model (LLM)-based multi-agent systems (MAS). The core motivation addresses critical limitations of existing MAS: (1) aimless hypothesis generation without rational constraints, (2) weak links between hypotheses and evidence, and (3) poor generalization across scientific domains.


The authors evaluate PiFlow across three distinct scientific tasks—nanohelix geometry optimization (NHO), molecular bio-activity prediction (MBO), and superconductor critical temperature optimization (SPO)—using high-fidelity surrogate models. Results show PiFlow outperforms baselines (ReAct, MPO, Vanilla MAS) by 73.55% in exploration efficiency (AUC) and 94.06% in solution quality (SQ) on average.

**Strengths:**

The writing is easy to follow.

The paper introduces new concepts: PiFlow addresses a fundamental gap in existing MAS: the lack of explicit integration of scientific principles into discovery workflows. By treating principles as actionable, iteratively refinable "guides" (rather than static domain knowledge), the work moves beyond "black-box" LLM reasoning to a more interpretable, scientific-first paradigm. This aligns with the needs of experimental research, where hypotheses must be grounded in causal mechanisms (not just correlations).

This paper proposes the practical plug-and-play modules. The Plug-and-Play design is a key strength for real-world adoption. The authors successfully integrate PiFlow with ChemToolAgent (a chemistry-focused MAS) to discover high-bioactivity molecules (pChEMBL = 5.90) without modifying the agent’s architecture (Appendix K). Additionally, PiFlow reduces token consumption by up to 27% vs. Vanilla MAS (Section 5.3), addressing cost concerns for long-horizon scientific tasks.

**Weaknesses:**

[1] The paper defines scientific principles as "foundational concepts or patterns articulated in natural language" (Definition 3.1) but leaves critical details unresolved:
Origin of initial principles: The authors mention principles may come from domain experts or LLMs, but how are LLM-extracted principles validated for accuracy? For example, LLMs are prone to hallucinations—does PiFlow include a mechanism to filter or correct flawed initial principles (beyond iterative refinement via evidence)?
Principle representation: The paper uses text embeddings to measure principle novelty (for exploration), but how are embeddings aligned with scientific relevance? A semantically distant principle could be irrelevant (e.g., a biology principle in superconductivity research)—does PiFlow incorporate domain boundaries to avoid such errors? The authors are suggested to add a small study comparing expert-provided vs. LLM-extracted initial principles (e.g., how hallucinated principles affect convergence) and clarify how embedding-based novelty scoring is constrained to domain-relevant principles.


[2] While PiFlow outperforms general-purpose baselines (ReAct, MPO), it does not compare to domain-specific state-of-the-art (SOTA) methods for the three tasks:
For NHO: How does PiFlow compare to physics-informed neural networks (PINNs) or inverse design frameworks (e.g., Xie et al., 2023b, cited in the paper) that explicitly model chiral optics?
For MBO: How does it stack against drug discovery tools like DeepChem or generative models (e.g., GFlowNets) optimized for molecular design?

**Questions:**

For the Plug-and-Play integration with ChemToolAgent: Did PiFlow require any domain-specific prompt engineering (e.g., adjusting how principles are phrased for chemistry tasks)? If so, how does this affect its claim of "domain agnosticism"?

---

> ### Author Response · Authors · 2025-11-21
>
> We thank the reviewer for highlighting the **novelty of our principle-aware paradigm** and the practical value of our **Plug-and-Play design**. We appreciate the opportunity to clarify how PiFlow’s **information-theoretic framework**inherently addresses validity and generalization.
>
> &nbsp;
>
> ### **[W1] Does PiFlow include a mechanism to filter or correct flawed initial principles (beyond iterative refinement via evidence)? How does it deal with irrelevant principles?**
>
> ---
>
> **PiFlow is the filter. The Min-Max optimization acts as an endogenous validity constraint.**
>
> We clarify that PiFlow moves beyond "black-box" reasoning by grounding principles in experimental feedback. It does not require an external filter because the correction mechanism is mathematically embedded in the objective function (Eq. 1):
>
> - **On one hand, we have theoretical guarantee:** The PiFlow objective combines **Exploration** (novelty) and **Exploitation** (performance $y_k$).
>     - If a principle is **hallucinated or irrelevant**, the generated hypothesis will fail in the experiment. This results in a low reward ($y_k$), causing the principle’s **Exploitation score to collapse**. Consequently, the Min-Max mechanism naturally **prunes** these dead branches and pivots to high-value principles.
> - **On the other hand, we have empirical proof:** We provide concrete evidence of this robustness in **Appendix P**. Instead of random hallucinations, we deliberately initialized PiFlow with **"Expert-Incorrect" principles** (misleading physics rules).
>     - **Result.** As shown in Table 10, the system detected the low exploitability of these principles within about 10 iterations. PiFlow successfully abandoned the flawed priors and recovered to converge on high-quality solutions (about 70% SQ).
>
> **Conclusion:** This demonstrates that PiFlow is robust to hallucination not via external filtering, but through **active, evidence-based correction**.
>
> &nbsp;
>
> ### **[W2] Compare to domain-specific SOTA (PINNs, GFlowNets, etc.).**
>
> ---
>
> **PiFlow is like the strategist (for uncertainty reduction across various domains), while PINNs/GFlowNets are instruments.**
>
> We clarify the positioning of PiFlow to address why general baselines (ReAct, MPO, BO) are the correct comparison group:
>
> - **Distinct roles of domain-specific SOTA and PiFlow:**
>     - **SOTA models like PINNs/GFLowNets act as specialized solvers, whereas PiFlow serves as a high-level strategic agent.**
>     - **Result.** We validate this hierarchy in **Appendix K** by integrating PiFlow with **ChemToolAgent** in a plug-and-play manner, and compare with the ChemToolAgent alone:
>         - Without any architectural modifications, PiFlow orchestrated the domain agent to improve molecular bio-activity from a pChEMBL of **2.80 (43% SQ)** **to 5.90 (91% SQ)** **within just 6 hypothesis-testing**.
>         - However, ChemToolAgent alone is unable to adhere to the chemical optimization gradients: in Iter 2, it fortuitously identified a promising **Phenylpropanoic acid scaffold** (Ibuprofen, **pChEMBL=3.53**), then the agent irrationally discarded this high-value aromatic structure in Iter 3, jumping to a structurally unrelated **aliphatic fatty ester** (Ethyl 2-methylbutyrate, **pChEMBL=2.11**) and even lower in the rest iterations (same 6 budget with PiFlow-ChemToolAgent), causing the bio-activity to collapse.
>     - Crucially, **PiFlow demonstrated strategic reasoning by identifying a promising "Quinazoline" scaffold (Iter 5)** and switching its command from Explore to Refine, thereby guiding the hypothesis to exploit the high-value region rather than searching blindly.
>
> **Conclusion.** As we clarified, PiFlow is designed to **control** domain-specific tools, not to replace them.
>
> &nbsp;
>
> ### **[Q] Did PiFlow require domain-specific prompt engineering?**
>
> ---
>
> **No. We utilized a unified prompting strategy (for scientific discovery by principles).**
>
> We confirm that **zero domain-specific prompt engineering** was used, reinforcing our claim of **Domain Agnosticism**.
>
> - **Implementation.** We used **the exact same generic system** prompts across all three disparate domains (Nanohelix, Molecules, Superconductors).
> - **Why it works:** PiFlow operates on the **mathematical structure** of discovery, not semantic nuances:
>     - **Exploration:** Computed via **embedding distances** (agnostic to text content).
>     - **Exploitation:** Computed via **normalized numerical regret** (agnostic to physical units).
>
> **Conclusion.** This implementation confirms that the logic of PiFlow is transferable without manual adaptation.
>
> &nbsp;
>
> We hope these responses clarify the robustness of our validation mechanism and the positioning of our framework.

---

### Official Review · Reviewer_oopF · 2025-10-30

**Soundness:** 3
**Presentation:** 3
**Contribution:** 3
**Rating:** 6
**Confidence:** 3

**Summary:**

The paper introduces PiFlow, a principle aware strategic layer for multiagent scientific discovery. The system situates an LLM based MAS inside a Hypothesis and Validation loop and adds PiFlow as a plug and play “director” that, at every step, selects and steers high potential scientific principles to guide hypothesis generation and experimentation. PiFlow formalizes principle selection as a minimax objective that balances regret minimization with information gain, operationalized via dynamic scoring and three actions, Explore, Validate, and Refine, issued to the MAS.

**Strengths:**

* Clear Definition 3.1 and a concrete mechanism that uses structured principles.
* Min–Max trade-off between exploitation (regret) and exploration (mutual information), plus sublinear regret with empirical alignment plots.
* The paper’s motivation is well-justified, and the study has clear research value.

**Weaknesses:**

* All evaluations rely on surrogate functions (no wet-lab / ab-initio verification in the loop). These risks reward hacking and mis-calibration of information gain. Please quantify surrogate uncertainty and show robustness when the validator is misspecified or noisy.
* Lacks sensitivity analyses for exploration weight, principle-set size and quality (expert vs. LLM-extracted), and the Explore/Validate/Refine thresholds that partition principles by potential.
* Experiments ban external search and fix the base LLM (Qwen-Max); add results with at least one other LLM to test backbone-robustness and with retrieval enabled to assess interaction with literature grounding.

**Questions:**

* What changes when swapping the base LLM (e.g., GPT-4-class) or enabling literature retrieval? Provide at least a small-scale study.
* Consider adding one more strategic baseline and reporting significance tests for AUC.
* How are the exploitation and exploration terms computed from T_t in Algorithm 1?

---

> ### Author Response · Authors · 2025-11-21
>
> We thank the reviewer for the constructive feedback and the positive assessment. We value insight regarding surrogate risks and generalization. To address these, we conducted **5 new sets of experiments**  (repeated 3 times, reporting mean/std). The results strongly reinforce PiFlow’s core contribution: acting as a robust, principle-aware strategic layer.
>
> &nbsp;
>
> ### **[W1] Robustness against surrogate noise (addressing reward hacking).**
>
> ---
>
> **PiFlow demonstrates superior robustness, validating that scientific principles act as effective regularizers against noise.**
>
> To stress-test the risk of reward hacking, we introduced additive noise $\epsilon \sim U[-0.2, 0.2]$ (approx. 10% of effective range) to the NHO task surrogate.
>
> - **Result.** As baselines collapsed (chasing noisy gradients), PiFlow maintained high Solution Quality (SQ).
>
>     | Method | Noise Level | SQ (%) | AUC (%) |
>     | --- | --- | --- | --- |
>     | **PiFlow** | **0% (Clean)** | **76.82 $\pm$ 4.54** | **63.51 $\pm$ 11.18** |
>     | **PiFlow** | **10%** | **73.65 $\pm$ 2.54** | **61.82 $\pm$ 1.95** |
>     | MPO (Baseline) | 10% | 50.46 $\pm$  5.04 | 41.74 $\pm$  4.87  |
>     | Vanilla (Baseline) | 10% | 51.38 $\pm$ 2.88 | 42.64 $\pm$ 3.09 |
>     | ReAct (Baseline) | 10% | 50.00 $\pm$ 5.05 | 40.92 $\pm$ 5.04 |
>
> **Conclusion.** This confirms our core claim: PiFlow does not merely exploit numerical rewards; it steers discovery via logical validation. By forcing the agent to ground hypotheses in **Principles**, PiFlow resists overfitting to surrogate artifacts.
>
> &nbsp;
>
> ### **[W2] Sensitivity analysis on hyperparameters (principle set size, action threshold).**
>
> ---
>
> **PiFlow is insensitive to hyperparameter variations, confirming the stability of the Min-Max mechanism.**
>
> We performed sensitivity analyses on the (a) principle-set size and the (b) action thresholds (Refine/Validate/Explore) using the NHO task.
>
> **a. Analysis of principle set size |P|.** Varying $|P| \in \{3, 6, 9\}$ showed stable performance.
>
> - **Result.** While smaller sets ($|P|=3$, SQ 76.8%) focus search efficiently, larger sets ($|P|=9$, SQ 76.1%) maintain high quality despite increased exploration costs.
>
>
>     | Config | SQ (%) | AUC (%) |
>     | --- | --- | --- |
>     | init P = 3 | 76.82 ± 4.54 | 63.51 ± 11.18 |
>     | init P = 6 | 72.47 ± 9.34 | 49.34 ± 7.17 |
>     | init P = 9 | 76.13 ± 5.55 | 47.19 ± 4.10 |
>
> **b. Analysis of action thresholds test.** We tested Strict mode (Refine > 0.8, Validate > 0.5), Default (Refine > 0.7, Validate > 0.4), and Loose mode (Refine > 0.6, Validate > 0.3) settings.
>
> - **Result.** PiFlow is robust across a reasonable range. The "Default" and "Loose" settings perform similarly well. Performance only drops in the "Strict" setting because the system is forced to "Explore" too frequently, preventing it from refining good findings. This confirms the thresholds are not brittle magic numbers.
>
>
>     | Config | SQ (%) | AUC (%) |
>     | --- | --- | --- |
>     | Loose (0.6/0.3) | 70.65 ± 9.66 | 52.18 ± 13.68 |
>     | **Default (0.7/0.4)** | **76.82 ± 4.54** | **63.51 ± 11.18** |
>     | Strict (0.8/0.5) | 62.35 ± 11.48 | 46.85 ± 12.24 |
>
> **c. Analysis of exploration weight (equal to the $\lambda$ param).** We have already tested the effect of $lambda$ parameter in **Appendix Q.** The $\lambda$ critically governs the exploration-exploitation trade-off. The optimal choice is domain-dependent, i.e., larger values suit novel domains to prioritize exploration, while smaller values are better for well-defined ones to prioritize exploitation.
>
> **d. Quality effect of initial principles.** In **Appendix P**, we have already done the experiment of initial principles' quality by giving Expert-Correct and Exper-Incorrect. Governed by a principled exploration-exploitation trade-off, PiFlow not only capitalizes on valid initial knowledge but identifies, rejects, and systematically recovers from flawed guidance.
>
> **Conclusion.** PiFlow exhibits **structural stability** rather than parametric brittleness. The analyses confirm that the framework performs robustly across a wide range of hyperparameter settings, minimizing the need for extensive tuning when adapting to new scientific domains.

---

> ### Author Response · Authors · 2025-11-21
>
> ### **[W3 & Q1] What changes when swapping the base LLM or enabling literature retrieval?**
>
> ---
>
> **PiFlow generalizes across LLMs and effectively grounds external knowledge.**
>
> - **Analysis of backbone LLMs:** We have already validated PiFlow with models such as GPT-4o-mini and Claude-3.5 (see Appendix O):
>     - We extended our evaluation to include SOTA models like Claude-3.7-sonnet, GPT4.1-mini, and QwenMax. Quantitative results reveal that, while efficiency (AUC) is model-dependent—ranging from 38.60% (Claude-3.7) to 63.51% (QwenMax), **Solution Quality (SQ) benefits consistently from stronger reasoners.** Specifically, Claude-3.7-sonnet achieves the highest SQ of **78.50%**, outperforming the efficiency-optimized QwenMax (**76.82%**) by ∼1.7%.
>     - **Conclusion.** This confirms that PiFlow generalizes well, effectively converting superior base model capabilities into higher-quality optimization outcomes.
> - **Analysis of literature retrieval integration.** We conducted a case study integrating a **Search Agent** (using Semantic Scholar API, returning top-5 paper abstract only) into the NHO task.
>     - **Qualitative leap of ON/OFF retrieval:**
>         - OFF: without retrieval, agent hypothesizes parameter changes based on general intuition.
>         - ON (case study): with retrieval, the agent grounds hypotheses in specific literature.
>     - **Case study of search-added PiFlow (NHO task):**
>         - **The 1st retrieval provides one key variable.** The Search Agent retrieved papers linking "two-turn SiO2 nanohelices" to "circular dichroism control (directly related to our objective value)".
>         - **Utilization.** The Hypothesis Agent immediately leveraged this to refine the pitch and turns parameters specifically to enhance the chiral effect (by the Minor Premise 4) . This led to a rapid g-factor jump from **0.52 -> 1.58**. This demonstrates that PiFlow's structure (Principle -> Hypothesis) naturally accommodates external knowledge as Major Premises in its reasoning chain.
>         - **The 2nd retrieval provides theoretical support.** When the progress come to the study of `number of turns`, Search Agent finds literatures about *"two-turn SiO2 nanohelixes"* and circular dichroism: *"The circular dichroism described with g-factor is also presented here as a function of wavelength... leads to flexible control over the circular dichroism".*
>         - **Utilization.** Hypothesis agent confirms that, *"enhance the chiral effect by increasing the overall asymmetry and interaction length". While the outcome is incremental,* **1.585 -> 1.590,** this searching ensures the adjustements is appropriate.
>     - **Conclusion.** The triggered a logic-driven jump in objective value proves that, PiFlow's architecture naturally accommodates external knowledge.
>
> &nbsp;
>
> ### **[Q2] Consider adding one more strategic baseline and reporting significance tests for AUC.**
>
> ---
>
> **PiFlow significantly outperforms the end-to-end AI Scientist baselines.**
>
> We compared PiFlow against AI-Researcher [1] (NeurIPS 2025) and The AI Scientist-v2 [2]. PiFlow's targeted uncertainty reduction (Min-Max) proves superior to generic autonomous loops.
>
> | **Task** | **Method** | **SQ (%)** | **P-value (SQ)** | **AUC (%)** | **P-value (AUC)** |
> | --- | --- | --- | --- | --- | --- |
> | **NHO** | PiFlow | 76.82 ± 4.54 | - | 63.51 ± 11.18 | - |
> |  | The-AI-Scientist-v2 | 56.67 ± 5.79 | 0.0103 | 49.27 ± 1.84 | 0.1547 |
> |  | AI-Researcher | 53.12 ± 1.06 | 0.0091 | 46.45 ± 2.42 | 0.1122 |
> | **MBO** | PiFlow | 84.55 ± 29.63 | - | 64.57 ± 23.65 | - |
> |  | The-AI-Scientist-v2 | 62.47 ± 8.02 | 0.3250 | 36.32 ± 10.62 | 0.1630 |
> |  | AI-Researcher | 95.66 ± 7.45 | 0.5870 | 42.72 ± 15.56 | 0.2625 |
> | **SPO** | PiFlow | 34.85 ± 1.19 | - | 21.51 ± 2.80 | - |
> |  | The-AI-Scientist-v2 | 29.85 ± 2.68 | 0.0663 | 28.43 ± 3.13 | 0.0470 |
> |  | AI-Researcher | 25.69 ± 3.23 | 0.0277 | 16.36 ± 2.72 | 0.0842 |
>
> **Conclusion.** PiFlow achieves statistically significant improvement on NHO and SPO tasks compared to AI Researcher and The AI Scientsit-v2. On the MBO task, PiFlow remains competitive with higher efficiency (AUC 64.57 vs 42.72 and 36.32), though SQ variance is higher. This confirms PiFlow's advantage in complex, uncertainty-heavy environments.

---

> ### Author Response · Authors · 2025-11-30
>
> ### **[Q3] How are the exploitation and exploration terms computed from $T_t$ in Algorithm 1?**
>
> ---
>
> **Both terms are derived from the trajectory history** $T_t = \{\langle p_k, y_k \rangle\}_{k=1}^t.$
>
> - **Exploitation (regret):** Computed via the normalized empirical outcome $y_k$ of principles in history (favoring high-performing principles).
> - **Exploration (information gain):** Computed via cosine distance between the candidate principle's semantic embedding and the set of validated principles in $\mathcal{T}_t$ (favoring novelty). Algorithm 1 (Line 9) dynamically balances these to minimize cumulative regret while maximizing information coverage.
>
> &nbsp;
>
> We will include these additional ablation studies and significance tests in the final manuscript to further strengthen the empirical evaluation.
>
> &nbsp;
>
> ## Reference
>
> [1] Tang, J., et al. AI-Researcher: Autonomous Scientific Innovation. In Advances in Neural Information Processing Systems, vol. 38, 2025.
>
> [2] Yamada, Yutaro et al. “The AI Scientist-v2: Workshop-Level Automated Scientific Discovery via Agentic Tree Search.” ArXiv abs/2504.08066 (2025): n. pag.

---

### Official Review · Reviewer_VUQd · 2025-11-02

**Soundness:** 2
**Presentation:** 3
**Contribution:** 2
**Rating:** 4
**Confidence:** 4

**Summary:**

The paper proposes a novel framework, PiFlow, for principle-aware scientific discovery using multi-agent collaboration. PiFlow integrates strategic principle-guided exploration into the discovery process, which effectively balances hypothesis generation and experimental validation through a Min-Max optimization framework. The method is shown to outperform baselines across multiple domains, making it a significant contribution to the field of automated scientific discovery.

**Strengths:**

- The paper presents a solid theoretical foundation and detailed experimental results, showing the efficacy of PiFlow in improving discovery efficiency and solution quality.
- The idea of principle-aware scientific discovery is highly original. The method’s integration of Min-Max optimization for balancing exploration and exploitation offers a fresh approach to scientific inquiry.
- PiFlow offers substantial improvements in scientific discovery workflows, making it a potentially transformative tool for automating research in complex scientific domains. The robustness and adaptability of the method, as shown in various experiments, highlight its broad applicability.

**Weaknesses:**

- The paper mentions that agents can access historical information, but it does not explain how this information is presented to the agents. This is critical, as long system prompts combined with multiple iterations may lead to context issues. Additionally, since each agent has a different role and task, it is unclear whether the historical information available to each agent differs. This aspect requires clarification to understand how context is managed effectively across agents.
- The paper does not describe a precise mechanism for controlling the number of iterations during the discovery process. Without such a control, the system may operate inefficiently, especially in resource-intensive scientific discovery tasks where early termination could help avoid unnecessary computations. The lack of a stop condition or a method to control the loop depth limits PiFlow’s practical usability in real-world applications.
- PiFlow relies on predefined principles to provide suggestions for refinement, validation, or exploration. However, the paper does not provide a detailed discussion on how the system generates initial principles when no predefined principles are available at the start of the process.

**Questions:**

- Could you elaborate on the strategy for managing historical information within long system prompts? How is this information organized to prevent context collapse?
- Is there a way to control the number of iterations in the discovery process to ensure efficient exploration within a predefined time frame?
- How does PiFlow handle situations where no initial principles are available at the start of the discovery process?

---

> ### Author Response · Authors · 2025-11-21
>
> We sincerely thank the Reviewer for recognizing PiFlow’s **solid theoretical foundation,** **originality,** and **transformative potential.** We appreciate the constructive feedback on context management and practical efficiency.
>
> Below, we clarify how our system design inherently addresses these concerns, further validating the robustness of our **principle-aware framework**.
>
> &nbsp;
>
> ### **[W1] How historical information is managed and does it differ across agents?**
>
> ---
>
> **We manage historical information via two independent parts: (a) agent memory for MAS to iterate the hypothesis-testing and (b) persistent principle pool for PiFlow layer to guide the process.**
>
> **a. For MAS part, agent history differs per agent.**
>
> - As the MAS follows round-robin order, there is the risk of context collapse. To prevent this, each agent maintains an independent `MessageBuffer`, retaining the most recent $M=10$ messages. This ensures agents focus on the immediate logical flow without being overwhelmed by noise.
>
> **b. For PiFlow, as a plugin layer, it has its own memory.**
>
> - Crucially, our **principle-aware design** decouples *reasoning history* from *scientific insight*. While chat logs may be truncated, **generated principles and validation scores represent compressed, high-value knowledge** stored in a separate, persistent global pool.
> - As per **Algorithm 1 (Lines 14-23)**, even if an agent forgets a specific past conversation, PiFlow retrieves the refined principles from the global pool to guide the next step. This ensures the *logic chain* remains intact throughout the discovery process.
>
> With the design of decoupling, PiFlow only need to collect historical principles, hypotheses and outcomes to optimize the strategy (explore, refine or validate which temporary principle), and steers the learning process by giving guidance to MAS.
>
> &nbsp;
>
> ### **[W2] The strategies of iteration control / stop condition.**
>
> ---
>
> **In PiFlow, fixed iterations were for benchmarking; dynamic termination is naturally supported by our Min-Max convergence.**
>
> - **We fixed iteration for fair comparision.** We strictly fixed the process to 24 iterations solely to ensure a**standardized computational budget comparison** across all baselines.
> - **Realworld efficiency (stopping by condition).** In practical deployment, PiFlow's Min-Max optimization naturally drives the system toward convergence, enabling precise threshold-based stopping:
>     - **Result.** In the Molecular Bio-activity Optimization (MBO) task, PiFlow-MAS improved pChEMBL from 2.69 to **6.44 within just 4 effective iterations**, and surpassed the target (6.5) to reach **7.65 by the 6th iteration.** This demonstrates that with a practical stop condition, PiFlow achieves success using only **~25% of the allocated benchmark budget**.
>     - **Theoretical guarantee of convergence.** As detailed in **Section 5.4**, our Min-Max framework provides a theoretical guarantee that prevents infinite loops of low-quality exploration, ensuring resource efficiency even in the intensive tasks noted by the Reviewer.

---

> ### Author Response · Authors · 2025-11-21
>
> ### **[W3] How does PiFlow handle cases with no pre-defined principles?**
>
> ---
>
> **The Min-Max framework autonomously generates and self-corrects principles from scratch.**
>
> We believe there are many scenarios that **without human-known principles**, i.e., **no pre-defined principles** that LLMs can propose. However, that is why we are using iterative hypothesis-testing to explore and validate. We discuss two realworld situations below:
>
> - **If there are no pre-defined principles:**
>     - **Firstly, initialization for solving the Cold Start.** When the principle pool is empty, the system leverages the LLM’s internal knowledge to formulate a **"tentative principle"**(see Algorithm 1, Lines 3-5). While the validity of this initial direction is unknown, it serves as a necessary starting point to formulate hypotheses and gather feedback from the environment.
>     - **Secondly, self-correction via Min-Max.** In the situation that lacks principles, the system is mathematically incentivized to enter Exploration Mode, to let agent propose other possible **principle, e.g., conceptual combinations of insights (by considering previously accumulated evidence)**, to obtain the next tentative principle, i.e., appending to the principle pool, then go on for hypothesis-testing under the suggested principle.
> - **If initial principles are wrong but with pre-defined principles:**
>     - As demonstrated in **Appendix P (Figure 10)**, even when intentionally initialized with *incorrect* principles (SQ is about 10%), PiFlow successfully rejects the flawed guidance via experimental feedback and **recovers to ~70% SQ.** This confirms that **PiFlow does not rely on golden initial principles**; it is designed to explore them through the optimization loop.
>
> **Conclusion.** PiFlow uses principles for scientific discovery, making it a highly efficient system **to hypothesize** **direction by direction, rather than candidate by candidate**.
>
> &nbsp;
>
> We hope these clarifications demonstrate that the concerns regarding context and initialization are effectively handled by PiFlow's architectural design.

---

### Author Response · Authors · 2025-11-30
**Global Response (1/2)**

**We sincerely thank the reviewers for their insightful feedback, recognizing PiFlow’s solid theoretical foundation, originality, and transformative potential as a novel framework for autonomous scientific discovery.** We have actively incorporated feedback, conducting extensive evaluations that further validate PiFlow not only as a robust optimizer but as a **lightweight, theoretically grounded, and highly generalizable strategic layer** for agentic systems. We believe these updates comprehensively address all of the reviewers’ questions and clearly demonstrate high merit of our work.

&nbsp;

1. **Summary of rebuttal actions**

We focused on empirically validating the robustness, efficiency, and generalization capabilities of PiFlow. We conducted **5 new sets of experiments**, including sensitivity analyses (Appendix Q), noise robustness tests (Appendix Q), ablations on principles initialization (Appendix P), and **comparisons against the latest SOTA agentic baselines**, i.e., AI Researcher and The AI Scientist-v2 (refer to Section 5.1 and Appendix I).

&nbsp;

2. **Evaluation against latest strong baselines**

To address concerns regarding comparative performance (Reviewer oopF), we extended our evaluation to include `AI Researcher` (NeurIPS 2025) and `The AI Scientist-v2` (2025) as baselines (Reviewer oopF).

- **Results:** PiFlow achieves statistically significant improvement ($p-value < 0.05$) over these baselines on NHO and SPO tasks (Section 5.1 and Appendix I).
- **Significance:** In the MBO task, PiFlow demonstrates superior efficiency (AUC 64.57 vs. 42.72 for AI-Researcher), confirming that our Min-Max strategy effectively targets uncertainty reduction better than generic autonomous loops.

&nbsp;

3. **Comprehensive updates of manuscript and Appendix**

We have expanded the Appendix to incorporate these findings, reinforcing the position of PiFlow as a flexible and robust framework:

- **Appendix G (Mechanism clarification):**
    - We detailed the **decoupling of memory** (Agent History vs. Principle Pool) and provided theoretical and empirical proofs for **Min-Max convergence.** This addresses context management concerns and theoretically guarantees the system's efficiency (Reviewer VUQd).
- **Appendix J (Numerical baselines):**
    - We clarify that PiFlow uniquely solves the **semantic hypothesis search** problem, an area where numerical baselines (GA/RL) struggle, offering the flexibility to handle complex real-world constraints (Reviewer TzZQ).
- **Appendix K (Generalization):**
    - We updated the case study integrating PiFlow with **ChemToolAgent**. Results demonstrate that PiFlow acts as a **strategic layer** that guides domain-specific agents to success without requiring architectural modifications. This validates our claim that PiFlow is designed to **control and elevate** domain-specific tools (like PINNs/GFlowNets) rather than replace them (Reviewer bk9d).
- **Appendix L (Efficiency & Bias):**
    - Runtime profiling reveals the logic of PiFlow consumes only **~9.4%** of total inference time. Furthermore, we demonstrate how semantic principles act as **strategic regularizers**, preventing overfitting to biased datasets (Reviewer TzZQ).
- **Appendix O (LLM robustness):**
    - Benchmarks with **Claude-3.7-Sonnet**, **GPT-4.1-mini**, and **QwenMax** prove that PiFlow consistently converts stronger base-model reasoning into higher solution quality, confirming its model-agnostic nature (Reviewer oopF).
- **Appendix P (Initialization & Self-correction):**
    - In "Cold Start" and "Expert-Incorrect principles initialization" scenarios, PiFlow successfully **self-corrects**via experimental feedback. It recovers high solution quality even from flawed initial principles, proving it does not rely on "golden" priors (Reviewer VUQd, bk9d, TzZQ).
- **Appendix Q (Robustness & Retrieval):**
    - **Noise Ablation:** Experiments with surrogate noise ($\epsilon \sim$10%) demonstrate **exceptional robustness**. PiFlow relies on principle-aware optimization, effectively preventing the "reward hacking" common in baseline agents (Reviewer oopF).
    - **Sensitivity Analysis:** We validated the stability of PiFlow across varying principle set sizes and action thresholds in our Algorithm 1 (Reviewer oopF).
    - **Literature Retrieval:** We also added a case study of PiFlow with literature retrieval, demonstrating how PiFlow grounds hypotheses in external literature (Reviewer oopF).
- **Correction of a transcription error:** During the rebuttal phase, we carefully re-verified our experimental logs and noticed a transcription error in Table 1 (Section 5, line 332), the AUC value of PiFlow in MBO was incorrectly copied as `46.11 ± 16.25` but should be `64.57 ± 23.65`. We have corrected this in the revised PDF. **This correction does not affect the relative ranking of methods or the conclusion of the paper.** We apologize for this oversight.

---

### Author Response · Authors · 2025-11-30
**Global Response (2/2)**

4. **Conclusion**

The extensive experiments and rigorous analyses confirm that PiFlow is a new **systematic framework** for scientific discovery. It stands out as a **lightweight, highly robust, and easily integrable** solution that effectively addresses the field's core challenges:

1. **Aimless hypothesizing:** Addressed by our Min-Max framework, which statistically outperforms generic agent loops regarding uncertainty reduction.
2. **Disconnected evidence:** Addressed by our principle-aware architecture that maintains logical consistency via a decoupled memory.
3. **Limited generalization:** Addressed by a **domain-agnostic, Plug-and-Play design**. We validated this across three distinct domains and external tool/agent system integrations, PiFlow offers a **seamless pathway** to empower existing agent systems with rigorous scientific reasoning.

&nbsp;

We believe these additions comprehensively address the reviewers' comments and firmly establish the validity and impact of our work.

---

### Meta-Review · Area_Chair_jSYD · 2026-01-11

**Summary:**

This paper describes PiFlow, a “principle-aware” strategic layer for LLM-based multi-agent scientific discovery, formalized as a min–max objective balancing regret (exploitation) and information gain (exploration). Reviewers agreed the framing is interesting and the method is clearly motivated. However, across the reviews and discussion, several concerns remained:

All three domains are evaluated with surrogate validators. Even with the added noise tests, this remains the core limitation: the paper does not convincingly establish that the claimed “uncertainty reduction” and “principle grounding” translates beyond these surrogate loops, where reward hacking and domain misspecification are hard to rule out. This was the central concern of oopF and also TzZQ.

The paper also still does not engage with domain-specific SOTA approaches in a way that would persuade a broad ICLR audience that PiFlow provides an advance for each domain rather than a generic orchestration wrapper.

The authors argue the min-max loop will prune hallucinated/irrelevant principles via low reward, but that is still a surrogate-driven argument. The paper does not fully pin down when “principles” are genuinely explanatory constraints versus just another prompt-level heuristic that happens to correlate with the surrogate objective.

**Reviewer Concerns:**

Surrogate-only evaluation remains the main blocker (oopF, TzZQ): The rebuttal improves testing but does not change key limitation: there is no convincing demonstration that the proposed principle-aware uncertainty reduction is robust when the validator is meaningfully misspecified relative to the real scientific target.

Comparative evidence is not persuasive (oopF, bk9d, TzZQ): Added baselines are a step forward, but the statistical picture for AUC is mixed, and the narrative around gains is not consistently supported by the reported numbers. The lack of domain-SOTA comparisons remains a gap for the domain claims.

Conceptual sharpness of “principles” as more than prompting: The paper still reads, in places, as a sophisticated orchestration/prompting layer whose “principle” objects are evaluated mainly by surrogate reward and embedding distance. The work would benefit from stronger evidence that these principles function as stable, transferable constraints rather than convenient intermediate text artifacts.

**Reviewer Scores:**

Even if discussion raised two of the “4” scores to “5,” the paper would still sit in a borderline zone with unresolved core validation and evidentiary issues, which is not enough to recommend acceptance here.

---

### Decision · Program_Chairs · 2026-01-26

Reject